# Site-specific monoubiquitination downregulates Rab5 by disrupting effector binding and guanine nucleotide conversion

**Donghyuk Shin, Wooju Na, Ji-Hyung Lee, Gyuhee Kim, Jiseok Baek, Seok Hee Park, Cheol Yong Choi, Sangho Lee***

Department of Biological Sciences, Sungkyunkwan University, Suwon, Korea

**Abstract** Rab GTPases, which are involved in intracellular trafficking pathways, have recently been reported to be ubiquitinated. However, the functions of ubiquitinated Rab proteins remain unexplored. Here we show that Rab5 is monoubiquitinated on K116, K140, and K165. Upon co-transfection with ubiquitin, Rab5 exhibited abnormalities in endosomal localization and EGF-induced EGF receptor degradation. Rab5 K140R and K165R mutants restored these abnormalities, whereas K116R did not. We derived structural models of individual monoubiquitinated Rab5 proteins (mUbRab5s) by solution scattering and observed different conformational flexibilities in a site-specific manner. Structural analysis combined with biochemical data revealed that interactions with downstream effectors were impeded in $mUbRab5_{K140}$, whereas GDP release and GTP loading activities were altered in $mUbRab5_{K165}$. By contrast, $mUbRab5_{K116}$ apparently had no effect. We propose a regulatory mechanism of Rab5 where monoubiquitination downregulates effector recruitment and GDP/GTP conversion in a site-specific manner.

DOI: https://doi.org/10.7554/eLife.29154.001

*For correspondence:
sangholee@skku.edu

**Competing interests:** The authors declare that no competing interests exist.

## Introduction

Rab GTPases, the largest family of small GTPases, regulate various vesicular trafficking processes such as endocytosis, exocytosis, and ER to Golgi transport (*Hutagalung and Novick, 2011*). Rab GTPases are activated by GTP binding and inactivated by the hydrolysis of GTP to GDP (*Pfeffer, 2005*). This regulatory cycle of Rab is facilitated by other regulators. A GDP dissociation inhibitor (GDI) stabilizes GDP-bound Rab in the inactive state. Binding of a GDI displacement factor (GDF) to the GDI:GDP-Rab complex releases GDI, freeing GDP:Rab for subsequent activation. A guanidine nucleotide exchange factor (GEF) binds to GDP:Rab5 and catalyzes GDP release and GTP loading for Rab. The activated GTP-bound Rab exerts its effects by binding to various effector proteins. Inactivation of GTP:Rab is aided by a GTPase activating protein (GAP) (*Cherfils and Zeghouf, 2013*), rendering the Rab GTPase ready for another cycle.

Ubiquitination, a major post translational modification, is associated with various cellular processes, including vesicular trafficking (*Hochstrasser, 2009*; *Komander and Rape, 2012*). Both the number of ubiquitin moieties and their chain linkage types, such as polyubiquitination and monoubiquitination, can provide the molecular bases for the regulation of diverse cellular activities. Two representative polyubiquitination linkages have been studied extensively. K48-linked polyubiquitination guides the substrate to the proteasome, resulting in degradation of the substrate (*Komander and Rape, 2012*; *Hochstrasser, 2009*; *Akutsu et al., 2016*). By contrast, K63-linked polyubiquitination regulates cellular signaling pathways (*Bhoj and Chen, 2009*). Monoubiquitination modulates histone modification, virus budding, and endocytosis (*Hicke, 2001*). Although ubiquitination in endocytosis

is well-documented (*Piper et al., 2014*), little is known about ubiquitination of Rab GTPases, the master regulators of endocytic trafficking (*Hutagalung and Novick, 2011*; *Mizuno-Yamasaki et al., 2012*).

Evidence is mounting that ubiquitination can alter the functional consequences of small GTPases, which has been well documented for Ras GTPases. Site-specific monoubiquitination of K147 of Ras leads to activation of signaling by impairing its interaction with GAP (*Baker et al., 2013a*; *Baker et al., 2013b*; *Sasaki et al., 2011*). In contrast, monoubiquitination of K117 of Ras facilitates GDP release from Ras, independent of a GEF (*Baker et al., 2013b*). Ubiquitination of Rab has also been recently reported. HACE1, a HECT-type E3 ubiquitin ligase, ubiquitinates Rab6A, Rab8A, and Rab11A in conjunction with the β2-adrenergic receptor (β2AR) (*Lachance et al., 2014*). Ubiquitination of K145 of Rab11A activates Rab11A. Atypical serine ubiquitination through the formation of a phosphodiester bond independent of E1 and E2 enzymes has been reported for Rab (*Qiu et al., 2016*; *Bhogaraju et al., 2016*; *Bhogaraju and Dikic, 2016*). Rab33b is monoubiquitinated through this atypical serine ubiquitination by SdeA, a pathogenic effector protein of *Legionella pneumophila*. However, the molecular mechanisms by which ubiquitination of Rab regulates Rab function remain largely unexplored.

Rab5 regulates formation of early endosomes and maturation to late endosomes (*Hutagalung and Novick, 2011*). Three isoforms of mammalian Rab5 - Rab5a, Rab5b and Rab5c regulate endocytosis in a co-operative manner (*Bucci et al., 1995*). However, recent studies suggest that these three isoforms may have differential roles. Rab5a is required for EGF-induced EGFR degradation while Rab5b and Rab5c are not (*Barbieri et al., 2000*; *Chen et al., 2009*). Rab5c is involved in cell migration whereas Rab5a and Rab5b are not (*Chen et al., 2014a*). Proteomic studies have suggested that the Rab5 isoforms can be differentially ubiquitinated in cells (*Chen et al., 2014b*; *Wagner et al., 2011*; *Wagner et al., 2012*). Since Rab5a has been extensively studied for its involvement in endosomal fusion (*Bucci et al., 1992*; *Gorvel et al., 1991*; *Barbieri et al., 2000*; *Hoffenberg et al., 1995*; *Rybin et al., 1996*; *Stenmark et al., 1994*), we decided to focus on Rab5a for our studies (hereafter referred to as 'Rab5'). Here, we identified monoubiquitination sites of Rab5 in cultured cells. Moreover, a modified chemical conjugation method enabled us to produce pure monoubiquitinated Rab5 proteins (mUbRab5s) in large quantities. Structural information about the mUbRab5s obtained from small-angle X-ray scattering and biochemical studies revealed that mUbRabs have different functions according to the site of ubiquitination. This research sheds light on how ubiquitination can regulate the function of Rab GTPases.

## Results

### Rab5 undergoes monoubiquitination at multiple sites

To determine whether Rab5 is ubiquitinated in cells as proteomics studies have suggested, we performed ubiquitination assays using HEK 293T cells. Rab5 was predominantly monoubiquitinated when FLAG-Rab5 was overexpressed with HA-ubiquitin in HEK 293T cells (*Figure 1a*). Because Rab5 is not known to be ubiquitinated by certain E3 ubiquitin ligases that can ubiquitinate other Rab proteins, such as the β2AR-HACE complex (*Lachance et al., 2014*) and the bacterial effector protein SdeA (*Qiu et al., 2016*; *Bhogaraju et al., 2016*; *Bhogaraju and Dikic, 2016*), monoubiquitination of Rab5 is likely to be performed by an as yet unidentified E3 ligase(s). Monoubiquitinated Rab5 migrated in a gel as doublet bands, implying that monoubiquitination of Rab5 can occur at multiple sites. Appearance of doublet bands from monoubiquitination was previously noted for caveolin-1 (*Ritz et al., 2011*; *Kirchner et al., 2013*; *Hayer et al., 2010*). To explore the possibility that double bands might have been caused by prenylation of Rab5 (*Xu and Nagy, 2016*; *Wojtkowiak et al., 2011*), we treated cells with a geranylgeranyl transferase inhibitor (GGTI-198) and then conducted ubiquitination assays (*Figure 1b*). Double bands from the monoubiquitinated Rab5 remained unchanged in cells treated with GGTI-198, demonstrating that the double bands are the result of monoubiquitination of Rab5 at multiple sites. These double bands were not observed when a high percentage polyacrylamide gel (12%) was used (*Figure 1c,e–f*). Next, we examined whether ubiquitination is dependent on the nucleotide binding state of Rab5. Ubiquitination assays with GDP- and GTP-bound forms of Rab5 mutants (S34N and Q79L, respectively) revealed that the monoubiquitination of Rab5 is not dependent on the nucleotide binding state of Rab5 (*Figure 1c*).

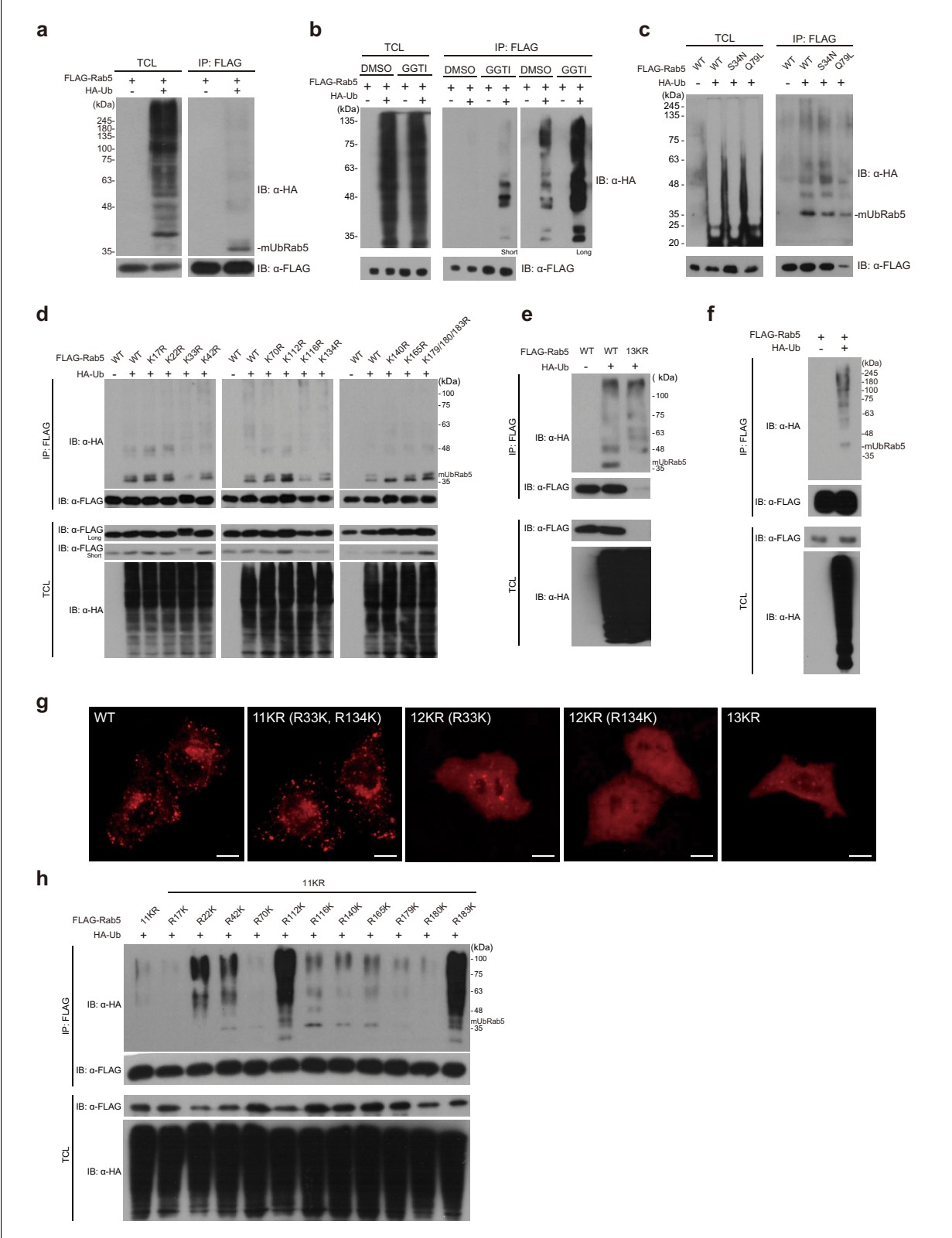

**Figure 1.** Rab5 is monoubiquitinated in the cellular environment. (a) Monoubiquitination of FLAG-Rab5 in HEK 293T cells. FLAG-Rab5 was co-transfected with HA-ubiquitin or HA vector and immunoprecipitated from total cell lysates (TCL). Immunoblotting was performed with the indicated antibodies. (b) Cells were treated with a geranylgeranyl transferase inhibitor (GGTI-298) or DMSO. After 24 hr of treatment, cells were harvested and subjected to ubiquitination assay. Short and long refer to exposure time during immunoblotting (c) GDP/GTP-bound mutants of Rab5 (S34N, Q79L)

*Figure 1 continued on next page*

*Figure 1 continued*

were examined by the ubiquitination assay (**d**) Each of 13 lysine (**K**) residues was mutated to arginine (**R**), and all single mutants were subjected to ubiquitination assay. (**e**) A 13KR (no lysine) mutant of Rab5 was generated and subjected to the ubiquitination assay. (**f**) FLAG-Rab5 was co-transfected with HA-ubiquitin or HA vector and subjected to ubiquitination assay in HeLa cells. (**g**) Immunofluorescence assay of 13KR, 12KR (13KR with R33K or R134K), and 11KR (13KR with R33/134K) mutants. FLAG-Rab5 WT and mutants were transfected into HeLa cells that were then stained with FLAG-mouse-IgG/rhodamine (red). Scale bar, 10 μm. (**h**) Ubiquitination assay of single lysine mutants in HEK 293T cells. The 11KR mutant was used as a negative control. Each of 11 arginine (**R**) residues was mutated back to lysine residues as indicated.

DOI: https://doi.org/10.7554/eLife.29154.002

The following figure supplement is available for figure 1:

**Figure supplement 1.** Structural and biochemical analyses of putative ubiquitination sites of Rab5.
DOI: https://doi.org/10.7554/eLife.29154.003

Proteomic studies reported the following ubiquitination sites for Rab5: K116 in human Rab5 (*Wagner et al., 2011*; *Chen et al., 2014b*) and K134 and K140 in mouse Rab5 (*Wagner et al., 2012*; *Chen et al., 2014b*) (*Table 1*). To verify whether those ubiquitination sites are monoubiquitinated and to examine if other lysine residues of Rab5 can be ubiquitinated, we mutated each of the 13 lysine (K) residues of Rab5 to arginine (R) and repeated the ubiquitination assay for each K-to-R mutant in HEK 293T cells. We found that the upper band of the doublet was completely abolished in the K140R mutant, while the monoubiquitination bands of Rab5 were not altered in the other K-to-R mutants (*Figure 1d*). This result suggests that K140 is a major monoubiquitination site of Rab5, and that at least two or more sites are responsible for the lower band.. To identify the ubiquitination sites on Rab5 comprehensively, we prepared a lysine zero mutant in which all 13 lysine residues of Rab5 were mutated to arginine (13KR). However, the 13KR mutant was not expressed in HEK 293T cells (*Figure 1e*).We reasoned that this was due to mutation of both K33 and K134 based on previous structural studies of Ras and EF-Tu GTPases (*Pai et al., 1989*; *Brünger et al., 1990*; *Jurnak, 1985*; *Berchtold et al., 1993*). Two lysine residues, the P-loop lysine in the GxxxGKS/T motif and the invariant lysine in the NKxD motif, are crucial because they form direct contacts with the guanine nucleotide. Similarly, for Rab5, K33 and K134 form direct contacts with both GDP and GTP molecules (*Figure 1—figure supplement 1b and c*, PDB ID: 1TU4 and 3MJH, respectively). K33, which sits in the nucleotide-binding site, performs two polar interactions with GDP/GTP: the Nζ atom and amide nitrogen interact with the Oβ and Oγ atoms of guanine nucleotides (2.7–3.4 Å for GDP; 2.8–3.5 Å for GTP; *Figure 1—figure supplement 1b and c*). K134 also contacts GDP/GTP through two interactions: Nζ atom interacts with O4' of the ribose ring (3.3 Å for GDP and 3.0 Å for GTP), and the amide nitrogen interacts with O6 of the guanine base (3.4 Å for both cases). Subsequently, we generated an 11KR mutant with intact K33 and K134 and examined its function by observing Rab5-positive endosome puncta in HeLa cells to visualize Rab5-positive endosomes more efficiently (*Miaczynska et al., 2004*; *Borg et al., 2014*; *Kajiho et al., 2003*). First, we checked whether Rab5 is monoubiquitinated in HeLa cells (*Figure 1f*). Rab5 was monoubiquitinated in HeLa cells, confirming that ubiquitination of Rab5 is conserved in the two cell lines. Whereas Rab5 puncta were not observed for the 13KR and the two 12KR mutants, puncta were formed as clearly in 11KR mutant cells as in WT cells (*Figure 1g*), suggesting that the functionality of the 11KR mutant is preserved. No monoubiquitinated Rab5 band was detected for the 11KR mutant in the ubiquitination assay (*Figure 1h*), suggesting that K33 and K134 are not responsible for monoubiquitination of Rab5. Considering that K134 was previously reported to be ubiquitinated by proteomic studies (*Chen et al., 2014b*; *Wagner et al., 2011*; *Wagner et al., 2012*), it is plausible that K134 may undergo polyubiquitination. Given the crucial role of K134 in the structural integrity of Rab5

**Table 1.** Putative ubiquitination sites of Rab5A based on published studies.

| Lysine site | Peptide from mass spectrometry | Predicted protein |
|---|---|---|
| 116 | RAKNWVKELQRQA | hRab5a (K116/215)(*Chen et al., 2014b*; *Wagner et al., 2011*) |
| 134 | IALSGNKADLANK | mRab5a (K134/215)(*Chen et al., 2014b*; *Wagner et al., 2012*) |
| 140 | KADLANKRAVDFQ | mRab5a (K140/215)(*Chen et al., 2014b*; *Wagner et al., 2012*) |

DOI: https://doi.org/10.7554/eLife.29154.004

(*Figure 1—figure supplement 1b and c*), we hypothesized that ubiquitination of K134 might be related to disruption/degradation of Rab5, thereby preventing us from observing ubiquitination of K134 in the current experimental conditions.

Using the 11KR mutant as a negative control, we introduced single R-to-K back-mutations in the background of the 11KR mutant and performed ubiquitination assays. Monoubiquitination bands were present for R116K, R140K, and R165K mutants in contrast to the 11KR mutant (*Figure 1h*). These results verified the previously known ubiquitination sites (K116 and K140) and revealed a novel ubiquitination site (K165). Interestingly, K165 corresponds to K147 of a Ras GTPase whose monoubiquitination activates Ras signaling (*Baker et al., 2013a*). Notably, R22K, R112K, and R183K mutants exhibited polyubiquitination patterns. Because no such polyubiquitination patterns were present in the WT (*Figure 1a–f*), these polyubiquitination patterns are likely artifacts from using the 11KR mutant as the background.

We analyzed these putative ubiquitination sites using the crystal structures of Rab5:GDP and Rab5:GTP (PDB ID: 1TU4 and 3MJH, respectively). None of the three lysine residues are located in the Rab5 switch regions (SW I and SW II; *Figure 1—figure supplement 1a*). In addition, all of the lysine residues are surface-exposed and accessible to ubiquitin (*Figure 1—figure supplement 1a*). Taken together, these findings indicate that, among the three lysine residues known to be ubiquitination sites of Rab5, K116 and K140 are monoubiquitinated, while K134 is not. We also identified K165 as a novel monoubiquitination site of Rab5.

## Monoubiquitination of K140 and K165 of Rab5 plays a negative role in the endocytic pathway

To elucidate the consequences of monoubiquitinated Rab5, we examined the effects of monoubiquitination on the formation of Rab5-positive puncta, because Rab5 itself is widely used as an early endosome marker protein (*Zhang et al., 2015*; *Liu et al., 2015*; *Alexopoulou et al., 2016*). We performed immunofluorescence assays in HeLa cells to examine the formation of endogenous Rab5-positivie puncta upon overexpression of ubiquitin (*Figure 2a–b*). Upon transfection of ubiquitin into HeLa cells, endogenous Rab5-positive puncta diffused to the cytosol and failed to co-localize with early endosome antigen-1 (EEA1) protein, another early endosome marker protein (*Figure 2a*). These results raise two possibilities: (i) monoubiquitinated Rab5 failed to be recruited to endosomal membranes or (ii) monoubiquitinated Rab5 disrupt Rab5-positive endosomal formation while it localized on the endosomal membranes. To distinguish between these two possibilities, we conducted a fractionation assay followed by immunoprecipitation to enrich for mUbRab5 in cytosolic and membrane fractions (*Figure 2c*). We observed that the monoubiquitinated Rab5 was predominantly localized in the membrane fraction. These results prompted us to hypothesize that monoubiquitinated Rab5 could disrupt Rab5-positive endosomal formation while it was localized on the membrane. Together with the immunofluorescence assay results (*Figure 2a–b*), these findings strongly suggest that ubiquitination of Rab5 has negative effects on the regulatory cycle of Rab5.

Next, we investigated whether the three putative ubiquitination sites (K116, K140, and K165) of Rab5 are involved in the endocytic pathway. We conducted EGF-induced EGF receptor (EGFR) degradation assays to examine the effects of monoubiquitinated Rab5 on the endocytic trafficking pathway (*Hoeller et al., 2006*; *Balaji et al., 2012*; *Smith et al., 2013*). EGFR was gradually degraded in EGF-treated HeLa cells, and complete degradation was observed after 90 min of EGF treatment (*Figure 2d* upper-left panel). Overexpression of Rab5 apparently facilitated EGF-induced EGFR degradation, as judged by earlier complete degradation after 60 min of EGF treatment (*Figure 2d* upper-right panel). Interestingly, overexpression of ubiquitin and Rab5 attenuated the degradation of EGFR, supported by the retention of some EGFR after 90 min of EGF treatment (*Figure 2d* middle-left panel). These results are consistent with the disruption of endosomal localization of Rab5 under ubiquitin overexpression (*Figure 2a–b*). Thus, ubiquitination of Rab5 is likely to play a negative role in the Rab5-mediated endocytic pathway. When the K140R mutant was overexpressed with ubiquitin EGFR degradation was completely restored and no attenuation was observed and almost all of the EGFR were degraded after 90 min of EGF treatment (*Figure 2d* lower-left panel), while the other mutants (K116R and K165R) still possesses such attenuation. This result suggests that ubiquitination of K140 of Rab5 has a crucial role in downregulating the Rab5-mediated endocytic pathway. To examine the effects of the three putative ubiquitination sites in the endosomal localization of Rab5, we conducted immunofluorescence assays. Consistent with the results obtained using

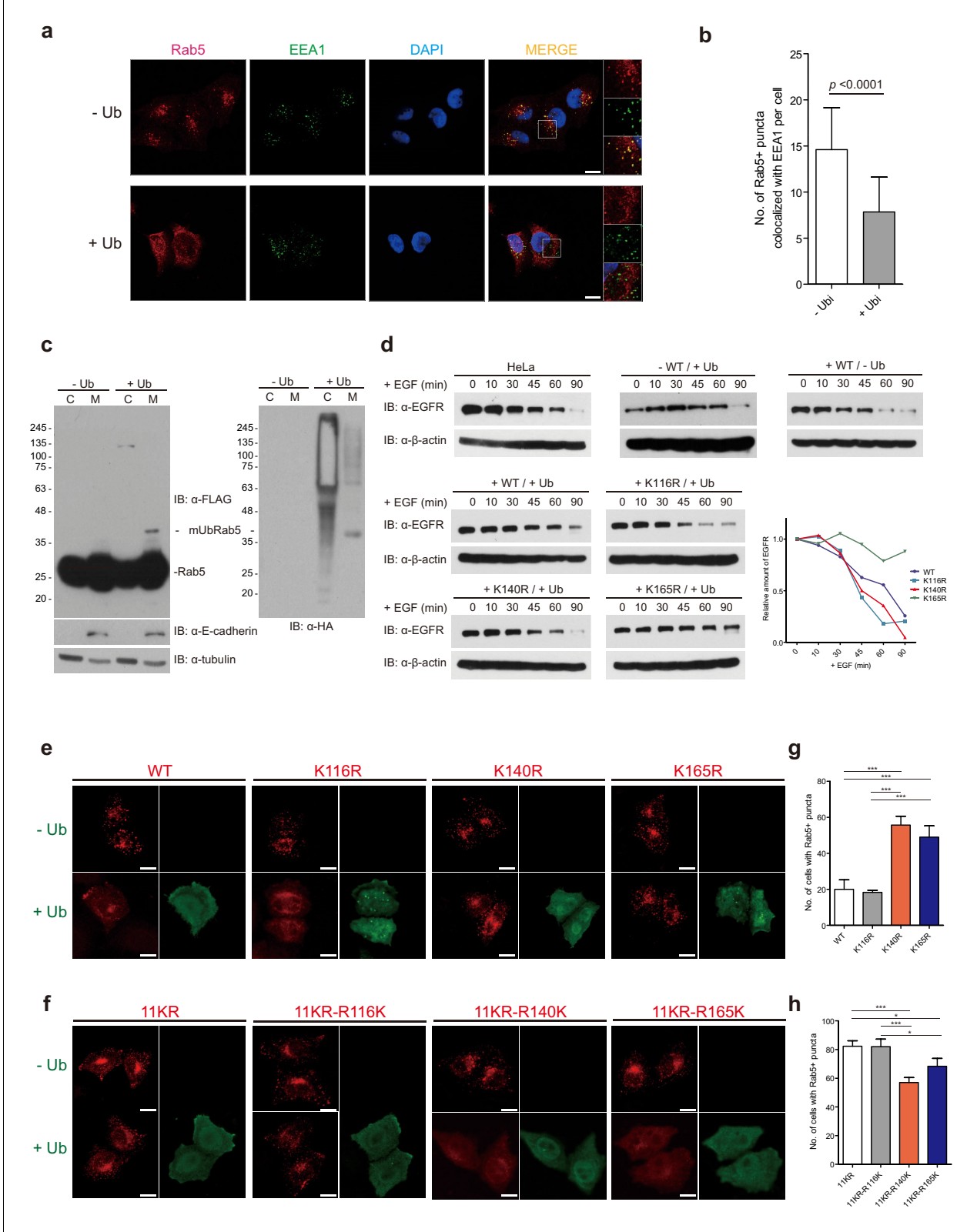

**Figure 2.** Endosome delocalization of Rab5 through overexpression of ubiquitin. (a) HeLa cells were transfected with HA-ubiquitin or HA vector. Endogenous Rab5 and EEA1 were immunostained with Rab5-mouse-IgG/rhodamine (red) and EEA1-rabbit-IgG/Alexa-fluor-488 (green). Images were obtained by confocal microscopy. Scale bar, 10 μm. (b) Quantification of immunofluorescence results in b. Number of Rab5 positive puncta co-localized with EEA1 is counted per cell (n = 45). p-value is calculated by t-test. (c) Fractionation assay. Cytosol and membrane fractions were subjected to
*Figure 2 continued on next page*

*Figure 2 continued*

immunoprecipitation and analyzed by immunoblotting as indicated. (**d**) EGF-induced EGF receptor (EGFR) degradation was monitored in HeLa cells. Cells were treated with EGF, harvested at the indicated time points, and analyzed by immunoblotting. (**e**) FLAG-Rab5 WT and K-to-R mutants were transfected with or without HA-ubiquitin into HeLa cells as indicated. (**f**) FLAG-Rab5 K11R and indicated 11KR/single R-to-K mutants were transfected with or without HA-ubiquitin into HeLa cells. Cells were immunostained with FLAG-mouse-IgG/Rhodamine (red) and HA-rabbit-IgG/Alexa-fluor-488 (green). Images were obtained by confocal microscopy. Scale bar, 10 μm. (**g, h**) Quantification of immunofluorescence results in f and g, respectively. Number of cells with Rab5-positive endosomes were counted from 100 cells transfected with ubiquitin. Data are presented as mean ± S.D ($n = 3$, *$p < 0.1$, ***$p < 0.001$, by one-way ANOVA).

DOI: https://doi.org/10.7554/eLife.29154.005

endogenous Rab5 (*Figure 2a–b*), Rab5-positive endosomal puncta were abolished when Flag-Rab5 WT was co-transfected with HA-ubiquitin (*Figure 2e,g*). Similarly, no puncta were observed for the K116R mutant upon co-transfection with HA-ubiquitin (*Figure 2e,g*), indicating that monoubiquitination of K116 does not affect Rab5-positive endosomal formation. By contrast, Rab5-positive endosomal puncta were detectable when either K140R or K165R mutants were co-transfected with HA-ubiquitin (*Figure 2e,g*). These observations suggest that monoubiquitination of K140 or K165 inhibits the Rab5-positive endosomal formation. To further investigated, we performed immunofluorescent assays using Rab5-11KR and three single-K-mutants (11KR-R116K, 11KR-R140K, and 11KR-R165K). Rab5-11KR mutant still displayed Rab5-positive endosomal puncta when it was co-transfected with HA-ubiquitin (*Figure 2f*). 11KR-R116K also showed Rab5-positive endosomal formation consistent with the previous finding that ubiquitination of K116 does not have an adverse effect on Rab5 function (*Figure 2e–h*). However, no endosomal puncta were observed for either 11KR-R140K or 11KR-R165K mutants upon co-transfection with HA-ubiquitin (*Figure 2f,h*). These results confirm the inhibitory role of monoubiquitination of either K140 or K165. These observations indicate that Rab5 monoubiquitination has functional consequences opposite those of Ras monoubiquitination (*Baker et al., 2013a*; *Baker et al., 2013b*): monoubiquitination of Rab5 on either K140 or K165 disrupts its function, whereas that of Ras on either K117 or K147 activates it. Taken together, our data demonstrate that monoubiquitination of K140 and K165 downregulates Rab5-mediated endocytic pathway, establishing monoubiquitination at specific sites of Rab5 as inhibitory signals, while monoubiquitination of K116 has no apparent effect.

## Modified chemical ubiquitination with iterative ubiquitin addition to obtain fully monoubiquitinated Rab5 protein

To understand how monoubiquitinated Rab5s negatively affect the Rab5 regulatory cycle at molecular level, we undertook biochemical studies of monoubiquitinated Rab5s. Structural, biophysical, and biochemical characterization of monoubiquitinated Rab5 proteins requires a large quantity of sample of the highest purity. Among the previously reported methods for generating ubiquitin-protein covalent linkages (*Merkley et al., 2005*; *McGinty et al., 2008*; *Freudenthal et al., 2010*; *Virdee et al., 2011*; *Rösner et al., 2015*), we adopted a chemical conjugation method where a disulfide bond is formed in place of the isopeptide bond, using K-to-C mutants of Rab5 (K116C, K140C and K165C) and the C-terminal G-to-C mutant of ubiquitin (G76C). A known problem with existing chemical conjugation methods is the incompleteness of conjugation. Because Rab5 and monoubiquitinated Rab5s have similar molecular masses and isoelectric points (25 kDa and pI 7.81 for Rab5; 34 kDa and pI 7.72 for monoubiquitinated Rab5), the presence of unmodified Rab5 would prevent separation of the two species by either size exclusion chromatography or ion exchange chromatography. Therefore, we modified the published methods (*Merkley et al., 2005*; *Baker et al., 2013a*) to overcome incomplete monoubiquitination of Rab5 by introducing iterative ubiquitin[G76C] addition (*Figure 3a*). First, we altered four native cysteine residues in Rab5 to prevent non-specific disulfide bond formation: two cysteine residues on the surface of Rab5 were mutated to serine (C19S and C63S), and four C-terminal residues, including two cysteine residues (CCSN; residues 213–216), were removed, inspired by a previous study (*Baker et al., 2013a*). GEF assays confirmed that the engineered Rab5 Del_Cys (C19S/C63S/Δ213–216) had the similar functionality as WT Rab5 (*Figure 3—figure supplement 1a,b*). The production of chemically conjugated monoubiquitinated Rab5 (hereafter referred to as 'mUbRab5') at a specific lysine residue by iterative addition of ubiquitin[G76C] at regular intervals was monitored by SDS-AGE under reducing (R) and non-reducing (NR) conditions. Increments in

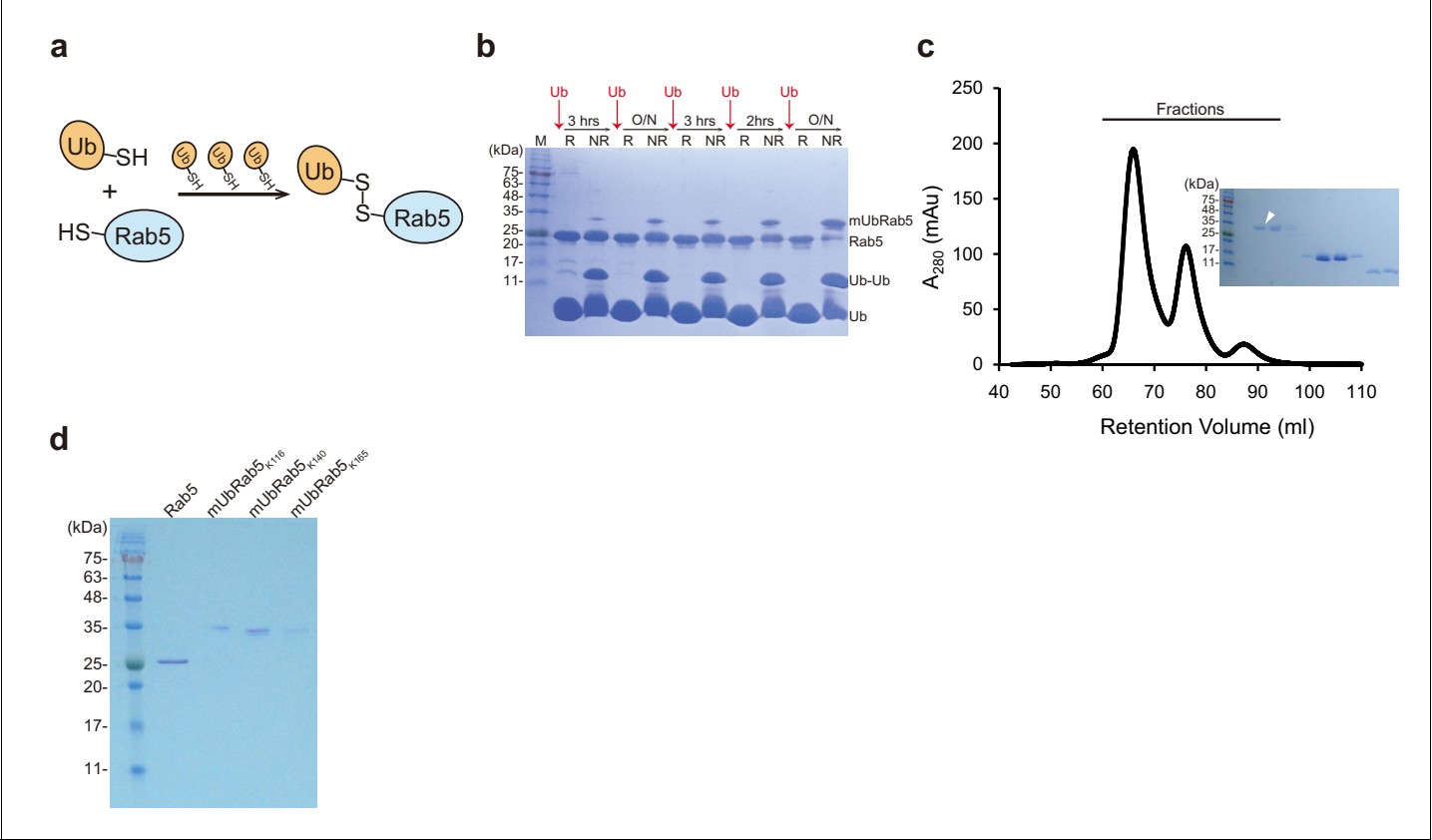

**Figure 3.** Chemical synthesis of mUbRab5s by iterative addition of ubiquitin$^{G76C}$. (a) Schematic diagram of the chemical ubiquitination of Rab5 by iterative addition of ubiquitin$^{G76C}$. (b) Production of mUbRab5$_{K140}$ was monitored by SDS-PAGE under reducing (R) or non-reducing (NR) conditions. Intervals between addition of ubiquitin$^{G76C}$ (Ub) are indicated. O/N, overnight. Data for the production of mUbRab5$_{K116}$ and mUbRab5$_{K165}$ are shown in *Figure 3—figure supplement 1c,d*. (c) A representative size exclusion chromatography chromatogram of the final reaction mixture for mUbRab5$_{K140}$ synthesis. (*Inset*) Fractions were analyzed by SDS-PAGE. White arrowhead indicates a fraction containing pure mUbRab5$_{K140}$. (d) Final products of chemically synthesized mUbRab5$_{K116}$, mUbRab5$_{K140}$, and mUbRab5$_{K165}$.

DOI: https://doi.org/10.7554/eLife.29154.006

The following figure supplement is available for figure 3:

**Figure supplement 1.** Functional comparison of Rab5 Del_Cys and WT, and chemical synthesis of mUbRab5$_{K116}$ and mUbRab5$_{K165}$.

DOI: https://doi.org/10.7554/eLife.29154.007

mUbRab5 and decrements in Rab5 band intensities were observed as more cycles of ubiquitin$^{G76C}$ addition were completed (*Figure 3b* and *Figure 3—figure supplement 1c,d*). Size exclusion chromatography successfully separated fractions containing only mUbRab5 (*Figure 3c* and *Figure 3—figure supplement 1c,d*). Finally, we obtained chemically synthesized mUbRab5$_{K116}$, mUbRab5$_{K140}$, and mUbRab5$_{K165}$ (*Figure 3d*). This simple modification resulted in significant improvement of mUbRab5 production and enabled us to perform structural and biochemical studies.

## Structural models of mUbRab5s derived by solution scattering

Despite the great interest in ubiquitinated proteins, very few structures of ubiquitinated proteins have been reported (*Merkley et al., 2005*; *Zhang et al., 2012*; *Freudenthal et al., 2010*). This might be due to the difficulties in purifying ubiquitinated proteins and the dynamics of the ubiquitin moiety; the ubiquitin moiety in ubiquitinated proteins has been reported to be able to adopt multiple conformations (*Baker et al., 2013a*; *Ye et al., 2012*). Consistently, structural simulations of ubiquitin moieties of mUbRas have predicted flexibility of these moieties (*Baker et al., 2013a*). Because of the inherent flexibility of the ubiquitin moiety of ubiquitinated proteins, we opted to employ small-angle X-ray scattering (SAXS). Three mUbRab5s (mUbRab5$_{K116}$, mUbRab5$_{K140}$, and mUbRab5$_{K165}$) were subjected to dynamic light scattering to evaluate whether the chemically

synthesized mUbRab5s were stable and monodisperse in solution. All three mUbRab5s were mono-disperse in solution (polydispersity <12%; *Figure 4—figure supplement 1a*). To conduct SAXS experiment mUbRab5$_{K116}$, mUbRab5$_{K140}$, and mUbRab5$_{K165}$ were concentrated up to 2.9, 7.3, and 2.6 mg/ml, respectively, and subjected to SAXS measurements (*Table 2*). All three mUbRab5s showed linear Guinier regions at low *q*, leading to determination of reliable values for a radius of gyration ($R_g$) (*Svergun, 1992*). To check whether the ubiquitin moieties of Rab5 are flexible, we per-formed flexibility analysis using the Porod-Debye method (*Rambo and Tainer, 2011*). SAXS curves from the three mUbRab5s did not have a plateau in the Porod-Debye plot, implying that all of the mUbRab5s were flexible in solution (*Figure 4—figure supplement 1b*). To understand the confor-mational dynamics of the mUbRab5s, we analyzed SAXS data using an ensemble-optimized method (*EOM*) (*Tria et al., 2015*). *EOM* was originally designed for multi-domain proteins with a flexible linker. Input files for *EOM* are a single linear protein sequence covering entire protein, atomic coor-dinates for each domain, and SAXS data (*Tria et al., 2015*). Because ubiquitinated proteins are not single polypeptides but branched ones, we generated the sequences and atomic coordinates of mUbRab5s as described in the Materials and methods. A pool of 10,000 independent models was generated based on the sequence and structural information from SAXS curve by *RANCH* (embed-ded in *EOM*, *ATSAS* package). Then, a genetic algorithm for the selection of an ensemble was per-formed by *GAJOE* with 100 times (embedded in *EOM*, *ATSAS* package). Finally, the best ensemble matched with the SAXS curve with lowest $\chi^2$ was selected. Distribution of conformational ensembles of mUbRab5$_{K140}$ was clearly distinct from those of mUbRab5$_{K116}$ and mUbRab5$_{K165}$. Two distinct populations of conformational ensembles were evident for mUbRab5$_{K140}$, with a radius-of-gyration ($R_g$) of 23 and 27 Å, respectively, and maximum distance ($D_{max}$) of 75 and 90 Å (*Figure 4b,e*). By contrast, mUbRab5$_{K116}$ and mUbRab5$_{K165}$ comprised a single population with an $R_g$ of 25 Å and

**Table 2.** SAXS data collection and analysis statistics.

|  | mUbRab5$_{K116}$ | mUbRab5$_{K140}$ | mUbRab5$_{K165}$ |
|---|---|---|---|
| *Data-collection parameters* | | | |
| Synchrotron beamlines | PAL-4C | | |
| Beam geometry | Capillary | | |
| Wavelength (Å) | 1.24 | | |
| Exposure time (sec) | 10 | | |
| Concentration range (mg/ml) | 0.4–2.9 | 0.9–7.3 | 1.0–2.6 |
| | | | |
| *Sample parameters* | | | |
| Polydispersity (%, by DLS) | 10.9 | 11.6 | 11.8 |
| | | | |
| *Structural parameters* | | | |
| $I(0)$ (cm$^{-1}$) [from Guinier] | 0.73 ± 0.01 | 4.29 ± 0.03 | 10.69 ± 0.05 |
| $R_g$ (Å) [from Guinier] | 25.54 ± 0.24 | 23.38 ± 0.34 | 27.23 ± 0.34 |
| $I(0)$ (cm$^{-1}$) [from $P(r)$] | 0.73 | 4.35 | 10.74 |
| $R_g$ (Å) [from $P(r)$] | 26.55 | 24.61 | 27.81 |
| $D_{max}$ (Å) | 86.56 | 83.51 | 93.69 |
| Porod volume estimate (Å$^3$) | 44428.4 | 32040.2 | 46685.0 |
| | | | |
| *Software employed* | | | |
| Primary data reduction | *RAW* | | |
| Dara processing | *PRIMUS* | | |
| Ensemble analysis | *EOM* and *FoXS MES* | | |
| Three-dimensional representations | *PyMOL* | | |

DOI: https://doi.org/10.7554/eLife.29154.016

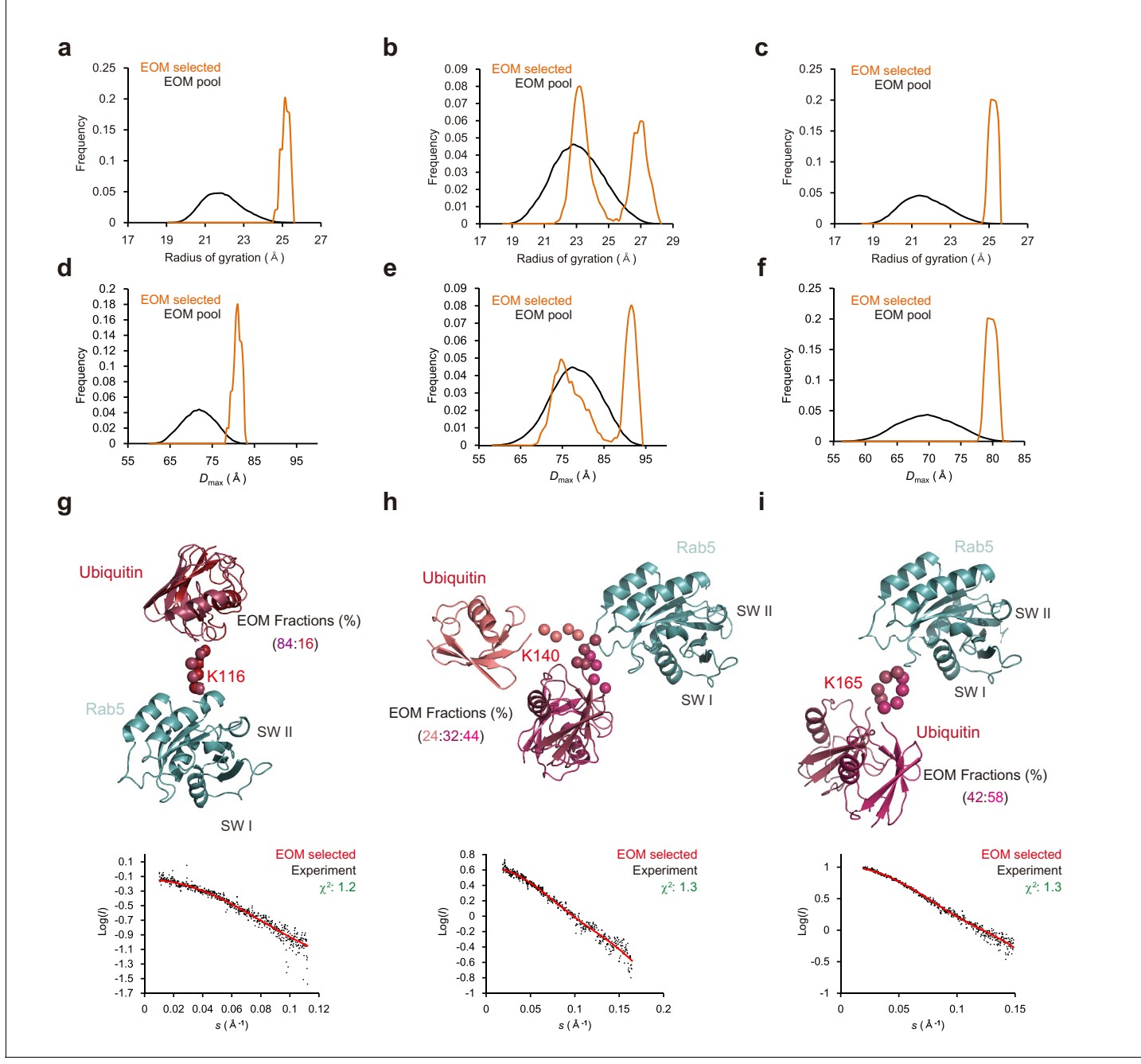

**Figure 4.** Ensemble models of mUbRab5s from small angle X-ray scattering (SAXS) obtained using the ensemble optimized method (*EOM*). (a, b, c) Radius of gyration ($R_g$) and (d, e, f) maximum inter-atomic distance ($D_{max}$) plots of the randomly generated pool (solid black) and *EOM*-selected model (orange) are shown for mUbRab5$_{K116}$, mUbRab5$_{K140}$, and mUbRab5$_{K165}$, respectively. (g, h, i) (*Upper*) Final ensemble models of mUbRab5$_{K116}$, mUbRab5$_{K140}$, and mUbRab5$_{K165}$. Each *EOM*-selected ensemble and switch I and II regions of Rab5 (SW I and SW II) are labeled. Rab5 is colored in cyan and ubiquitin is colored in series of red. (*Lower*) Raw data with the calculated SAXS curve for each model are plotted.

DOI: https://doi.org/10.7554/eLife.29154.008

The following source data and figure supplements are available for figure 4:

**Source data 1.** SAXS raw data and *EOM* results of mUbRab5$_{K116}$.

DOI: https://doi.org/10.7554/eLife.29154.013

**Source data 2.** SAXS raw data and *EOM* results of mUbRab5$_{K140}$.

DOI: https://doi.org/10.7554/eLife.29154.014

**Source data 3.** SAXS raw data and *EOM* results of mUbRab5$_{K165}$.

*Figure 4 continued on next page*

*Figure 4 continued*

DOI: https://doi.org/10.7554/eLife.29154.015

**Figure supplement 1.** Polydispersity and flexibility analyses of mUbRab5s.

DOI: https://doi.org/10.7554/eLife.29154.009

**Figure supplement 2.** Structural models of mUbRab5s with a GEF.

DOI: https://doi.org/10.7554/eLife.29154.010

**Figure supplement 3.** Structural models of mUbRab5s with effector proteins.

DOI: https://doi.org/10.7554/eLife.29154.011

**Figure supplement 4.** Structural models of mUbRab5s with a GAP.

DOI: https://doi.org/10.7554/eLife.29154.012

$D_{max}$ of 80 Å (*Figure 4a,d and and c,f*). These results strongly suggest that monoubiquitination at these three lysine sites has different functional consequences. *EOM*-derived structural models of mUbRab5s revealed specific orientations of the ubiquitin moiety. Three major conformations were derived for mUbRab5$_{K140}$ (*Figure 4h*) and two conformations for each of mUbRab5$_{K116}$ and mUbRab5$_{K165}$ (*Figure 4g,i*). This was in contrast to K147-monoubiquitinated Ras (mUbRas$_{K147}$) that exhibited different orientations of the ubiquitin moiety in the top 10 lowest-energy models from simulations (*Baker et al., 2013a*). The number of relative orientations of the ubiquitin moiety appears to be less than that of the major conformations derived: essentially, two orientations for mUbRab5$_{K140}$ and a single orientation each for mUbRab5$_{K116}$ and mUbRab5$_{K165}$. This indicates that the ubiquitin moiety on K140 is more flexible than those on K116 and K165. Our results suggest that the ubiquitin moieties on multiple lysine residues of Rab5 can assume different conformations, resulting in different modes of regulation. Similar to the molecular dynamics simulation of mUbRas (*Baker et al., 2013a*), none of the ubiquitin molecules from the *EOM* models were located proximal to Rab5. This raises the possibility that monoubiquitination of Rab5 can regulate its function by altering its interactions with partner proteins such as GEF, GAP, and downstream effectors.

To determine if monoubiquitination affected the interaction of Rab5 with its partners, we superimposed our mUbRab5 models with known complex structures of Rab5:Rabex-5 (a GEF), RabGAP-5 (a GAP), and Rabaptin5 and EEA1 (effector proteins). In the mUbRab5:Rabex-5 complex models obtained by superpositioning of mUBRab5 and the Rab5:Rabex-5 structure (PDB ID: 4DQU) (*Zhang et al., 2014*), none of the ubiquitin moieties of mUbRab5s appeared to clash with Rabex-5 (*Figure 4—figure supplement 2*). We speculated that the ubiquitin moieties might not interfere with activities related to a GEF, such as GDP release and GTP loading. Next, we generated structural models of mUbRab5:Rabaptin5 and mUbRab5:EEA1 by superposition on Rab5:Rabaptin5 and Rab5:EEA1 structures (PDB IDs: 1TU3 and 3MJH, respectively). The ubiquitin moieties of mUbRab5$_{K116}$ and mUbRab5$_{K165}$ were far from both Rabaptin5 and EEA1 (*Figure 4—figure supplement 3*). However, the switch I (SWI) region of Rab5 was juxtaposed with the flexible ubiquitin moiety of mUbRab5$_{K140}$, raising the possibility that a flexible ubiquitin moiety could cause a conformational change in the SWI. Because this conformational change of SWI is required for activation of Rab5 through interactions with downstream effector proteins, potential interruption of the SWI by mUbRab5$_{K140}$ could have a severe impact on the Rab5 regulatory cycle. Finally, we generated a structural model for mUbRab5:RabGAP-5 by superposition to a known Rab:GAP structure (PDB ID: 4HLQ). The ubiquitin moiety of mUbRab5$_{K116}$ did not clash sterically with RabGAP-5 (*Figure 4—figure supplement 4a*). Both mUbRab5$_{K140}$ and mUbRab5$_{K165}$ seemingly bring the ubiquitin moieties close to RabGAP-5 (*Figure 4—figure supplement 4b,c*), suggesting that neither mUbRab5$_{K140}$ nor mUbRab5$_{K165}$ would have enhanced interactions with RabGAP-5. Taken together, our structural analyses of mUbRab5s with partner proteins such as GEF, GAP, and effector proteins strongly suggest that the inhibitory effects of some mUbRab5s are highly likely to be caused by impedance of interactions with effector proteins.

## mUbRab5s retain GEF-mediated guanine nucleotide conversion

To dissect the inhibitory role of mUbRab5s at the molecular level, we examined whether monoubiquitination of Rab5 could interfere with guanine nucleotide conversion, because the major regulatory mechanism of Rab GTPase is the conversion between a GDP-bound state (inactive) and GTP-bound state (active) (*Novick, 2016*; *Mizuno-Yamasaki et al., 2012*; *Cherfils and Zeghouf, 2013*). Rab5

guanidine nucleotide exchange factor (Rabex-5) mediates the conversion from GDP to GTP for Rab5 (*Zhang et al., 2014*; *Thomas and Strutt, 2014*; *Aikawa et al., 2012*). We measured GDP dissociation and GTP association rates of the mUbRab5s in the presence and absence of a GEF (Rabex-5$_{132-393}$). Guanine nucleotide analogues 2'-(or-3')-*O*-(*N*-methyl-anthraniloyl) GDP (MANT-GDP) and MANT-GTP were used to detect the interaction between guanine nucleotides and Rab5. GDP release from either Rab5 or mUbRab5 was measured by monitoring the decrease in the amount of fluorescence from MANT-GDP when non-fluorescent GDP was added in the presence and absence of Rabex-5, a GEF (*Figure 5a,b* and *Table 3*). Both mUbRab5$_{K116}$ and mUbRab5$_{K140}$ showed similar GDP-release activities to WT Rab5 in the presence of Rabex-5, indicating that monoubiquitination of K116 and K140 does not interfere with the GEF function (*Figure 5a,b*). Notably, mUbRab5$_{K165}$ exhibited reduced GDP release in the presence of GEF, and the curve could not be fitted by a one-phase decay linear regression model. Next, GTP loading was monitored using MANT-GTP and Rabex-5. Again, mUbRab5$_{K116}$ and mUbRab5$_{K140}$ showed similar levels of GTP loading to Rab5 (*Figure 5c,d*). However, the fluorescence signal from mUbRab5$_{K165}$ decreased continuously after it reached a maximum. This observation indicates that monoubiquitination of K165 does not block GTP loading of Rab5, but can weaken the interaction between GTP and Rab5. Our data support the structural analyses that predicted that monoubiquitination of Rab5s would have no effect on the Rab5-GEF interaction (*Figure 4—figure supplement 2a–c*). Taken together, these findings suggest that monoubiquitination of Rab5 on K116 and K140 does not affect GEF activity, while monoubiquitination of K165 interferes with the GEF-mediated conversion cycle, as indicated by this protein's reduced response to GDP release and leakage during GTP loading.

## Monoubiquitination of K140 of Rab5 decreases its binding affinity to effector proteins

Our data demonstrate that endosome localization of Rab5 is downregulated by monoubiquitination of either K140 or K165, while monoubiquitination on K116 is not important for localization. Notably, we found that mUbRab5$_{K165}$ showed reduced GDP release activity and weakened GTP loading. However, the GEF activity of mUbRab5$_{K140}$ remained unchanged relative to WT. We then conducted downstream effector protein binding assays with Rab5 and mURab5s. Several studies have reported that monoubiquitination of a target protein can alter its interactions with other proteins (*Lin et al., 2016*; *Lau et al., 2015*; *Duan et al., 2016*). Previous studies of Ras monoubiquitination have revealed that monoubiquitinated Ras has different binding affinities to its effector proteins compared to WT Ras (*Sasaki et al., 2011*). This prompted us to examine the interactions between mUbRabs and effector proteins. We determined the dissociation constants ($K_d$) of Rab5 and mUbRab5s for two Rab5 effector proteins, Rabaptin5 and EEA1 (*Mishra et al., 2010*; *Zhu et al., 2004*), using bio-layer interferometry (*Figure 6*). Each of the Rab5 binding domains of Rabaptin5 and EEA1 (Rabaptin5$_{551-862}$ and EEA1$_{36-91}$) was highly purified as a GST fusion protein (*Figure 6a*). Rabaptin5, a well-characterized Rab5 effector protein, has been widely used to study biochemical and biophysical mechanisms involved in Rab5-mediated endocytosis (*Zhu et al., 2007*; *Zhu et al., 2004*; *Zhang et al., 2014*). Surprisingly, the $K_d$ value of mUbRab5$_{K140}$ for Rabaptin5 was five-fold higher than that of unmodified Rab5 (37.7 ± 4.9 μM *vs.* 8.0 ± 3.4 μM; *Figure 6b,c* and *Table 4*), indicating the reduced affinity of mUbRab5$_{K140}$ for Rabaptin5. However, mUbRab5$_{K116}$ and mUbRab5$_{K165}$ showed little change in the $K_d$ values: 8.0 ± 0.6 μM and 4.4 ± 0.2 μM, respectively. These results strongly suggest that monoubiquitination of K140 of Rab5 inhibits its binding to downstream effector proteins. Because effector proteins bind to Rab5 via similar binding modes (*Mishra et al., 2010*), it is plausible that mUbRab5$_{K140}$ can inhibit interactions with other Rab5 effector proteins. To evaluate this possibility, we determined the $K_d$ values of Rab5 and mUbRab5s for early endosome antigen-1 (EEA1), another well-known Rab5 effector protein. EEA1 recognizes phosphatidylinositol 3-phosphate-positive membranes and recruits Rab5 (*Christoforidis et al., 1999*; *Rubino et al., 2000*; *Murray et al., 2016*). mUbRab$_{K140}$ exhibited an approximately 20-fold increase in $K_d$ for EEA1 compared to unmodified Rab5 (108.0 ± 22.0 μM *vs.* 5.8 ± 0.3 μM, respectively; *Figure 6d,e* and *Table 4*). The $K_d$ value of unmodified Rab5 for EEA1 was similar to that reported previously (5.8 ± 0.3 μM *vs.* 2.4 ± 0.23 μM) (*Mishra et al., 2010*). mUbRab5$_{K116}$ showed a modest increase in $K_d$ value for EEA1 (18.4 ± 0.8 μM), while mUbRab5$_{K165}$ showed virtually no change (4.7 ± 0.5 μM) in $K_d$ value relative to that of unmodified Rab5. These data are consistent with the structural models, which suggested a possible clash of mUbRab$_{K140}$ with effector proteins but no

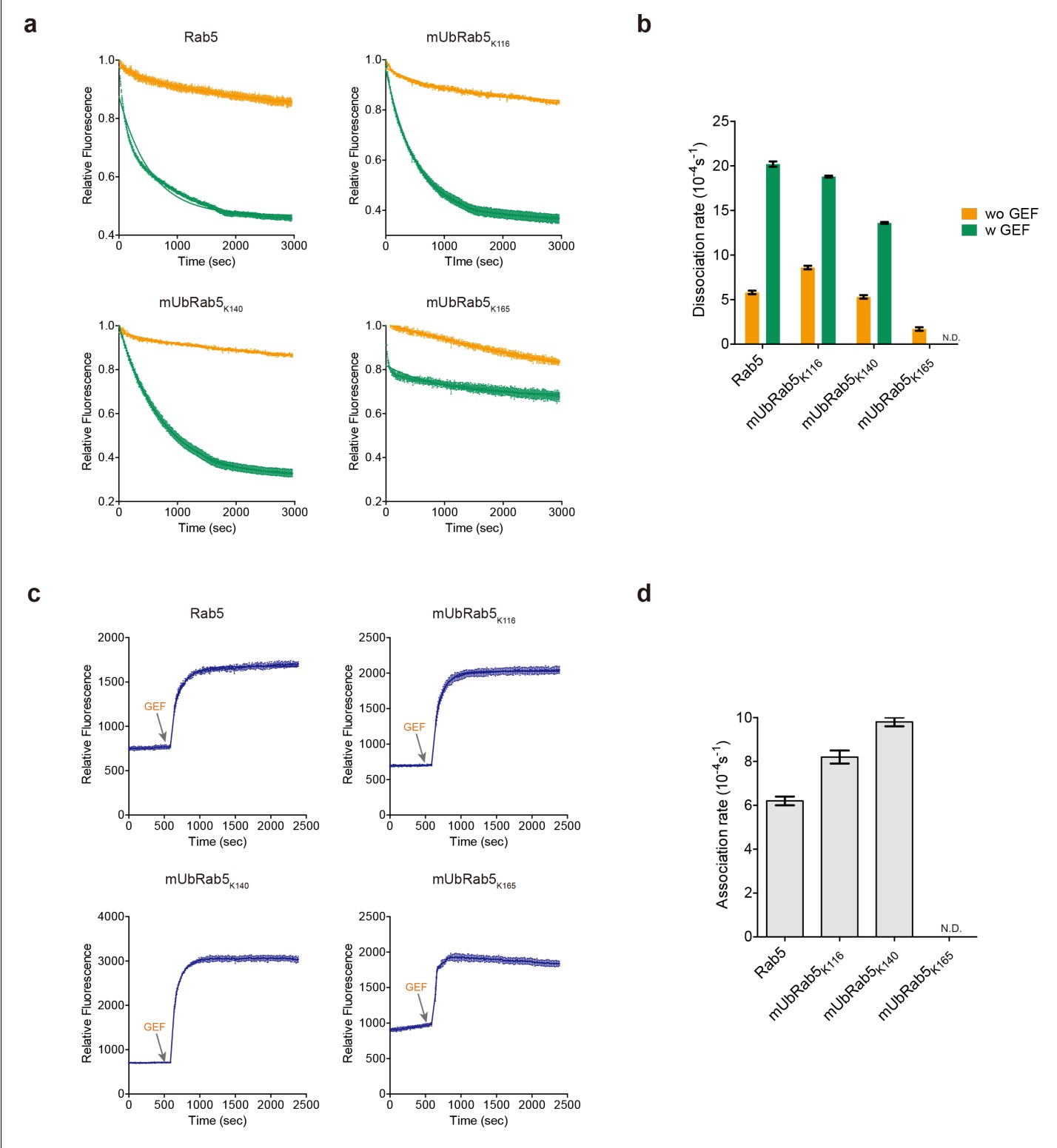

**Figure 5.** Intrinsic and GEF-mediated GDP release and GTP loading activities of mUbRab5s. (**a**) MANT-GDP release assay with (green) or without a GEF (orange; Rabex-5$_{132-393}$). Raw data and fitted one-phase exponential dissociation curves are shown. (**b**) Dissociation rates of MANT-GDP with or without the GEF are shown as mean ± standard deviation (S.D.; $n$ = 3). (**c**) MANT-GTP loading was monitored by fluorescence emission at different time points. GEF was added to the reaction mixture to facilitate the GTP loading at the indicated time (gray arrow). (**d**) Association rates of MANT-GTP with the GEF are shown as mean ± S.D. ($n$ = 3). Raw data and fitted one-phase exponential association curves are shown. (**b, d**) Dissociation and

*Figure 5 continued on next page*

*Figure 5 continued*

association rates of mUbRab5$_{K165}$ with GEF were not determined (labeled 'N.D.") because the data could not be fitted to one-phase exponential dissociation and association curves.

DOI: https://doi.org/10.7554/eLife.29154.017

The following source data is available for figure 5:

**Source data 1.** Raw data of GDP release assay for Rab5 and mUbRab5s.

DOI: https://doi.org/10.7554/eLife.29154.018

**Source data 2.** Raw data of GTP loading assay for Rab5 and mUbRab5s.

DOI: https://doi.org/10.7554/eLife.29154.019

such clash for mUbRab$_{K116}$ and mUbRab$_{K165}$ (*Figure 4—figure supplement 2d–i*). Our results collectively establish that monoubiquitination of K140 of Rab5 disrupts its interactions with down-stream effector proteins, while monoubiquitination of K116 only has a minor effect on these interactions.

## Discussion

Rab GTPases have recently been revealed to be ubiquitinated (*Lachance et al., 2014*; *Qiu et al., 2016*). However, the functional roles of ubiquitinated Rab GTPases have not been explored yet. Here, we demonstrated that Rab5 is monoubiquitinated at K116, K140, and K165 in two different cell lines (HEK 293T and HeLa). We also obtained structural ensemble models of the mUbRab5s by solution scattering. While the ubiquitin molecule on mUbRab5$_{K140}$ was flexible, it was not flexible from the other two mUbRab5s. mUbRab5$_{K140}$ shows decreased binding affinity to downstream effectors such as Rabaptin5 and EEA1, but its GEF-mediated GDP release and GTP loading activities were not changed. Through GDP release and association assays, we found that mUbRab5$_{K165}$ had altered binding to GDP/GTP. Based on our data, we propose a molecular mechanism for how site-specific monoubiquitination of Rab5 acts as an inhibitory signal (*Figure 7*). The canonical regulatory mechanism of Rab5 involves a GEF- and GAP-mediated GDP/GTP conversion cycle and interaction between activated Rab5 and downstream effector proteins (*Figure 7a*). Monoubiquitination of K116 of Rab5 does not interfere with any part of the functional cycle (*Figure 7b*). By contrast, monoubiquitination of either K140 or K165 of Rab5 negatively regulates the functional cycle by either reducing the interaction of Rab5 with downstream effector proteins (K140 monoubiquitination; *Figure 7c*) or altering the intrinsic GDP/GTP conversion cycle (*Figure 7d*).

Our proposed model for the role of monoubiquitination in the functional cycle of Rab5 is clearly distinct from the case of Ras (*Baker et al., 2013a*), where monoubiquitination interferes with GAP activity, leading to activation of Ras signaling. Therefore, monoubiquitination seems to play different roles in Rab and Ras GTPases: monoubiquitination in Rab5 serves as an inhibitory signal, whereas it serves as a stimulatory or activating signal in Ras. It will be interesting to determine the ubiquitination sites of other Rab GTPases and to investigate how ubiquitination alters the intrinsic functions of these proteins. Proteomics studies to screen for ubiquitination sites of Rab5 have also indicated that other Rab GTPases are ubiquitinated. As Rab5 monoubiquitination is different from Ras monoubiquitination, other Rab GTPases might be regulated differently by site-specific ubiquitination. For these

**Table 3.** GDP dissociation and GTP association rates for Rab5 and mUbRab5s.

| | GDP dissociation rate $\pm$ SD[*] ($10^{-4}$ s$^{-1}$) | | GTP association rate $\pm$ SD ($10^{-4}$ s$^{-1}$) |
|---|---|---|---|
| | Without GEF | With GEF | |
| Rab5 | 5.8 $\pm$ 0.2 | 20.2 $\pm$ 0.3 | 6.2 $\pm$ 0.2 |
| mUbRab5$_{K116}$ | 8.6 $\pm$ 0.2 | 18.8 $\pm$ 0.1 | 8.2 $\pm$ 0.2 |
| mUbRab5$_{K140}$ | 5.3 $\pm$ 0.2 | 13.6 $\pm$ 0.2 | 9.8 $\pm$ 0.2 |
| mUbRab5$_{K165}$ | 1.7 $\pm$ 0.2 | ND[†] | ND[†] |

* SD, standard deviation

† ND, not determined.

DOI: https://doi.org/10.7554/eLife.29154.020

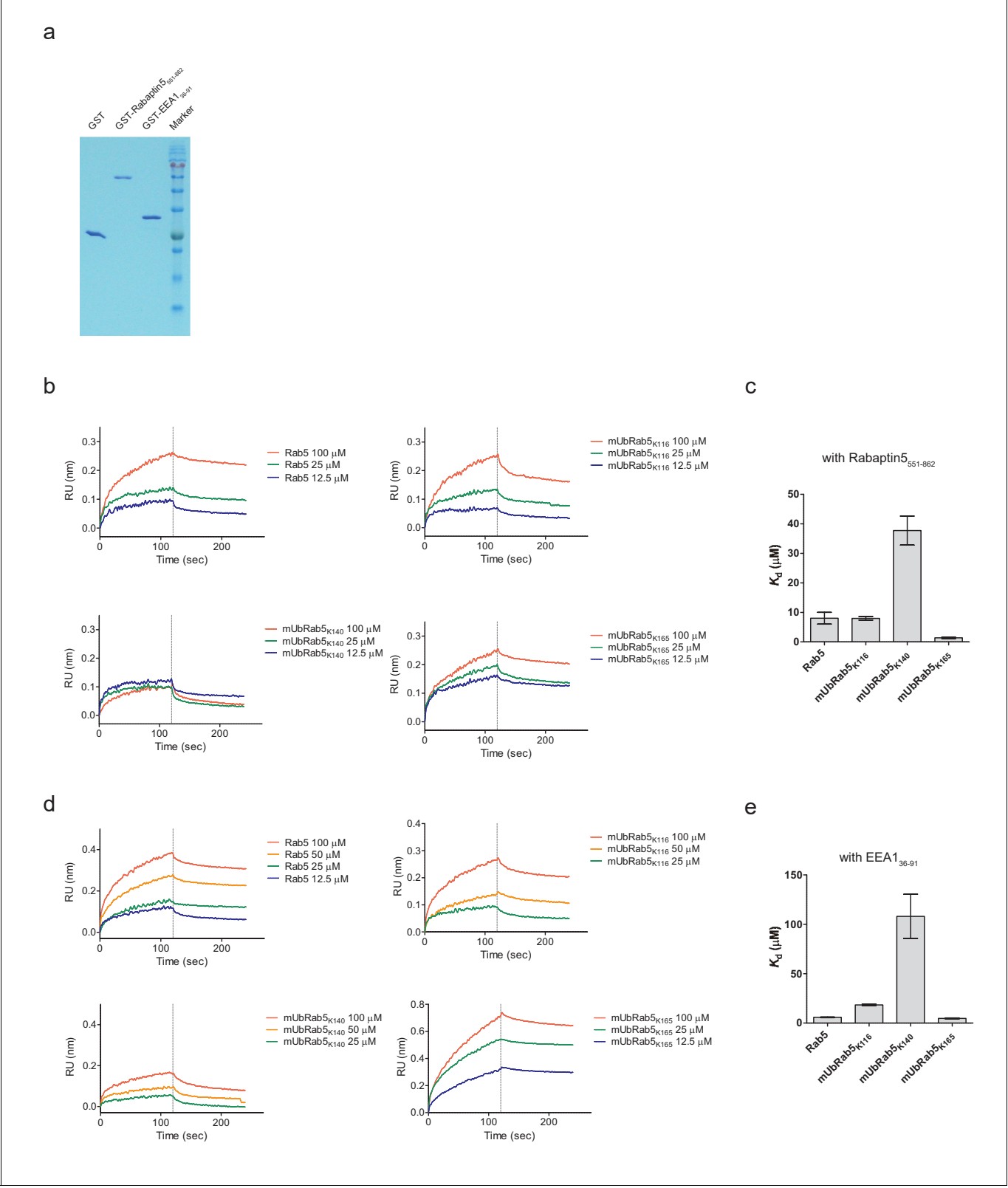

**Figure 6.** Impairment of the interaction of mUbRab5$_{K140}$ with effector proteins (Rabaptin5 and EEA1). (**a**) Highly purified GST-tagged Rabaptin5$_{551-862}$, GST-EEA1$_{36-91}$, and GST proteins were prepared. (**b, d**) Bio-layer interferometry assay was performed to determine dissociation constants ($K_d$) for mUbRab5$_{K140}$ and either Rabpatin-5 or EEA1. Several concentrations of Rab5 or mUbRab5s were used as indicated. (**c, e**) Dissociation constants were calculated by global binding fitting. $K_d$ values are shown as mean ± S.E.M.

*Figure 6 continued on next page*

*Figure 6 continued*

DOI: https://doi.org/10.7554/eLife.29154.021

The following source data is available for figure 6:

**Source data 1.** Raw data of BLI assays of Rab5 and mUbRab5s with Rabaptin5.

DOI: https://doi.org/10.7554/eLife.29154.022

**Source data 2.** Raw data of BLI assays of Rab5 and mUbRab5s with EEA1.

DOI: https://doi.org/10.7554/eLife.29154.023

studies, the optimized chemical ubiquitination method described here will be useful. Recently, atypical serine ubiquitination was reported for Rab33b (*Qiu et al., 2016*; *Bhogaraju et al., 2016*; *Bhogaraju and Dikic, 2016*), but the underlying molecular mechanism has not been clarified. Our chemical ubiquitination method can be used to determine the functional role of serine ubiquitination in Rab GTPases. Further investigation is required to evaluate the diversity of roles of monoubiquitination among different small GTPase super families and/or among members of the same superfamily.

Advances in proteomic techniques have revealed the ubiquitination sites of a small number of Rab proteins. Analysis of published proteomics data revealed that 28 of 63 human Rab GTPases (including isoforms) are ubiquitinated in cells. When we compiled ubiquitination sites of Rab proteins reported by proteomic studies, we found that ubiquitination sites of Rab5 found in the current study - K116 and K140 - are located in non-conserved regions, while K165 is in the conserved G5 motif (xxxETSx'K') (*Figure 8a*). It is plausible that ubiquitination of non-conserved lysine residues in Rab proteins and possibly small GTPases in general exhibits different functionalities. We noticed ubiquitination of K33 and K134 (numbering based on the Rab5 sequence) in some Rab proteins; both of these residues are critical for structural integrity (*Figure 1e,g*, *Figure 1—figure supplement 1*). Analysis of Rab5 proteins from eukaryotic species revealed different degrees of conservation of the three ubiquitination sites discovered in this study: both K116 and K165 are strictly conserved from yeast to human, whereas K140 is less conserved (*Figure 8b*). K140 is replaced by isoleucine in *Drosophila* and glutamate in yeast, implying that K140 might be a point of functional differentiation in these organisms. In contrast to eukaryotic Rab5 proteins, the three ubiquitination sites were not conserved in Rab5 homologue from *Acanthamoeba polyphaga mimivirus* (APMV). Recent crystal structure of APMV Rab5 revealed that K165 of hRab5a is substituted to F152 in APMV Rab5 and plays a major role for binding to GTP molecule (*Ku et al., 2017*). Therefore, it seems that monoubiquitination of Rab5 is a subsidiary signal for its regulation in eukaryotic cells. It is also intriguing that ubiquitination of a conserved site can have different functional consequences in Rab and Ras GTPase proteins. Monoubiquitination of K165 of Rab5 has an inhibitory effect, whereas ubiquitination of the corresponding K147 of Ras has a stimulatory effect (*Baker et al., 2013a*; *Baker et al., 2013b*). Therefore, monoubiquitination appears to be a versatile modification tool.

Oxidative conjugation of a ubiquitin moiety to a cysteine residue of a target protein has been widely used for chemical synthesis of ubiquitinated proteins. Despite the relative simplicity of this method in comparison with enzymatic conjugation, oxidative conjugation often suffers from incompleteness of the process, rendering subsequent separation of the ubiquitinated protein from

**Table 4.** Binding kinetics of Rab5 and mUbRab5s to Rabaptin5$_{551\text{-}862}$ and EEA1$_{36\text{-}91}$.

| | Rabaptin5 | | | EEA1 | | |
|---|---|---|---|---|---|---|
| | $k_{on} \pm$ SEM* ($10^2$ M$^{-1}$s$^{-1}$) | $k_{off} \pm$ SEM ($10^{-3}$s$^{-1}$) | $K_d$ $\pm$SEM ($\mu$M) | $k_{on}$ $\pm$SEM ($10^2$ M$^{-1}$s$^{-1}$) | $k_{off} \pm$ SEM ($10^{-3}$s$^{-1}$) | $K_d$ $\pm$SEM ($\mu$M) |
| Rab5 | 6.6 ± 2.8 | 2.2 ± 0.2 | 8.0 ± 3.4 | 4.5 ± 0.2 | 2.6 ± 0.1 | 5.8 ± 0.3 |
| mUbRab5$_{K116}$ | 2.4 ± 0.1 | 1.9 ± 0.1 | 8.0 ± 0.6 | 1.9 ± 0.2 | 5.0 ± 0.2 | 18.4 ± 0.8 |
| mUbRab5$_{K140}$ | 13.9 ± 1.8 | 52.3 ± 1.4 | 37.7 ± 4.9 | 1.8 ± 0.4 | 19.7 ± 0.8 | 108.0 ± 22.0 |
| mUbRab5$_{K165}$ | 3.2 ± 0.1 | 1.4 ± 0.1 | 4.4 ± 0.2 | 1.9 ± 0.8 | 0.9 ± 0.1 | 4.7 ± 0.5 |

* SEM, standard error of the mean

DOI: https://doi.org/10.7554/eLife.29154.024

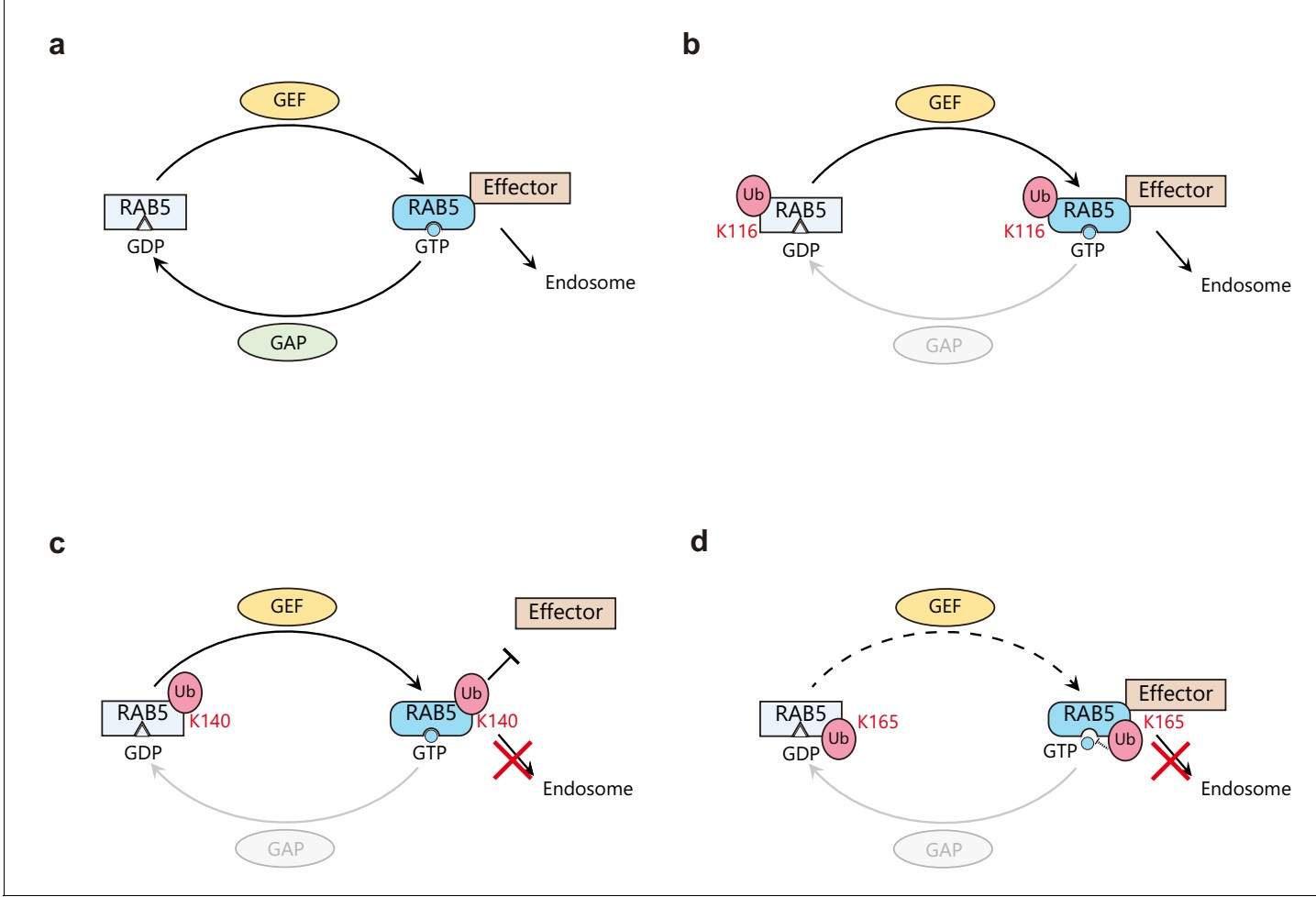

**Figure 7.** Proposed model of Rab5 regulation by site-specific monoubiquitination. (**a**) Canonical Rab5 regulatory cycle. Inactive GDP-bound Rab5 is activated by a guanine nucleotide exchange factor (GEF such as Rabex-5) through the exchange of GDP to GTP. The activated GTP-bound Rab5 interacts with various effector proteins to form early endosomes. Activated Rab5 is switched-off by a GTPase activating protein (GAP) through the hydrolysis of GTP to GDP. (**b**) Monoubiquitination on K116 has no effect on the regulatory cycle. Thus, mUbRab5$_{K116}$ is regulated in the same way as unmodified Rab5. (**c**) Monoubiquitination of K140 causes the ubiquitin moiety to impair the interaction of Rab5 with its effector proteins. Therefore, mUbRab5$_{K140}$ cannot form early endosomes. (**d**) Monoubiquitination of K165 alters the GDP/GTP conversion cycle by interfering with the activities of GEF, thus lowering affinity to GTP. mUbRab5$_{K165}$ cannot form early endosomes.

DOI: https://doi.org/10.7554/eLife.29154.025

unreacted protein very challenging (*Baker et al., 2013b*; *Merkley et al., 2005*). Such an inseparable mixture can obscure correct interpretation of biochemical and structural data. Here, we modified a pre-existing method by iterative addition of ubiquitin$^{G76C}$ to drive the chemical synthesis to completion. Our optimized method for chemical synthesis of ubiquitinated proteins will facilitate studies on the effects of protein ubiquitination at the molecular level when no identified E3 ubiquitin ligase is available or a large quantity of ubiquitinated protein is desired.

In this study, we found that monoubiquitination of Rab5 had an inhibitory effect on the regulatory cycle of Rab5. However, we did not find a stimulus signal or E3 ligase(s) responsible for monoubiquitination of Rab5. As result, only a small portion of Rab5 could be ubiquitinated when ubiquitin was overexpressed. These results suggest that ubiquitination of Rab5 occurs rarely in cells under basal conditions and may explain why the ubiquitination of Rab GTPases has not been studied until recently (*de la Vega et al., 2011*). Despite the small population of monoubiquitinated Rab5, the effects of monoubiquitination of Rab5 were strong. We observed dramatic failure of localization of Rab5 to the endosome and EGFR degradation pathway. This suggests that monoubiquitination of

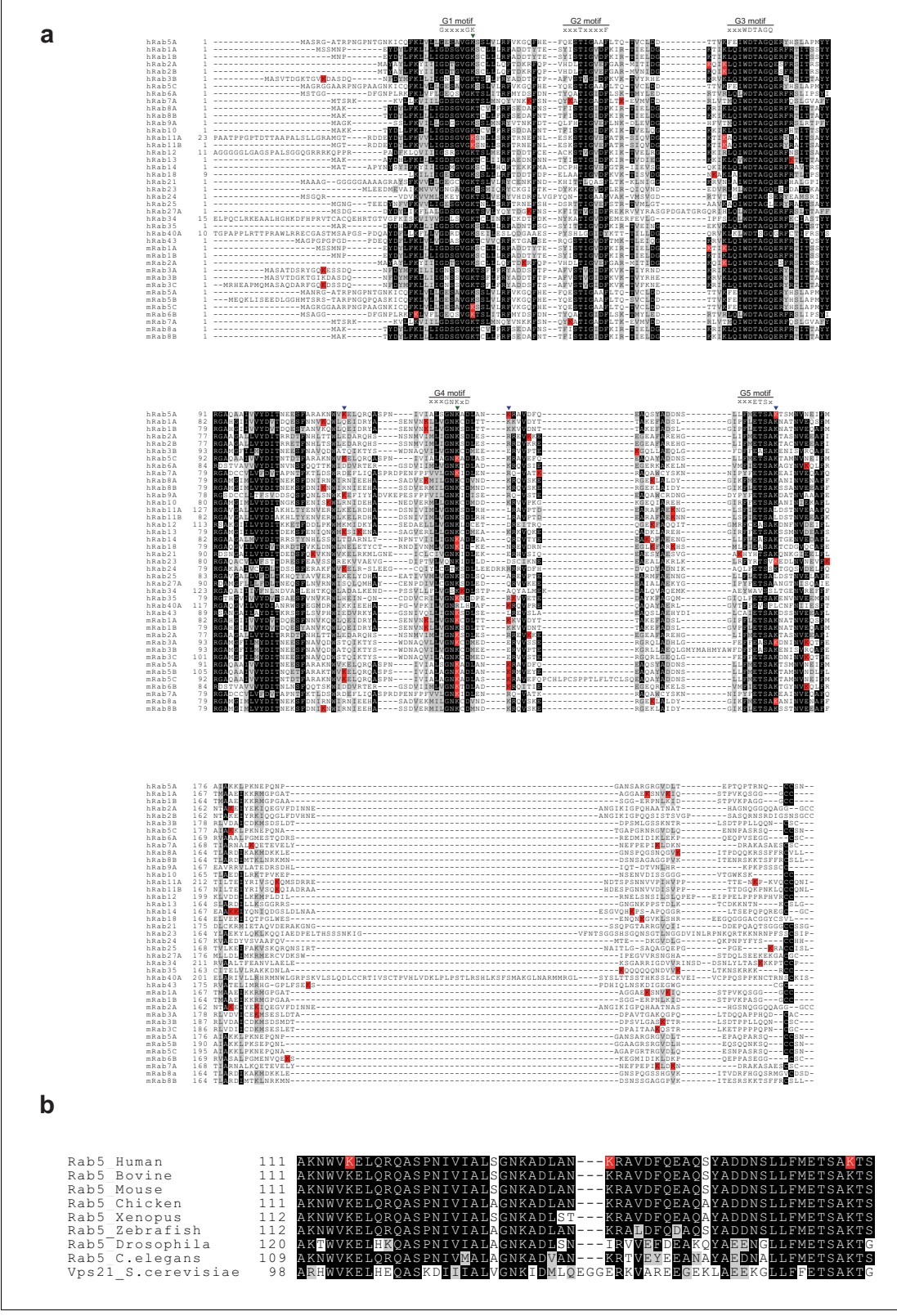

**Figure 8.** Multiple sequence alignment of human and mouse Rab GTPases reported to be ubiquitinated. Rab GTPases in the ubiquitination database (mUbSiDa [*Chen et al., 2014b*]) were obtained, and their sequences were aligned to human Rab5A (this study) using *Clustal X2* (*Larkin et al., 2007*). The graphical representation of the aligned sequences was prepared using *BoxShade (ExPASy server)*. Ubiquitination sites from proteomic studies are indicated with the letter 'K' in the red background (K). Ubiquitination sites and structural key lysine residues revealed from this study are marked as

*Figure 8 continued on next page*

*Figure 8 continued*

inverse triangles in black ( ▼ ) and green ( ▼ ), respectively. 'h' and 'm' refer to human and mouse, respectively. (**b**) Rab5 GTPases from different species were aligned to human Rab5A. Ubiquitination sites from this study are indicated with the letter 'K' on a red background (K).
DOI: https://doi.org/10.7554/eLife.29154.026

Rab5 is tightly regulated in the cell through specific E3 ligase or DUB enzymes. It will be of great interest to determine how monoubiquitination of Rab5 is regulated in cells.

## Materials and methods

### Plasmids and cell culture

The pCDNA-GFP-Rab5 clone was provided by Dr. Yoshikatsu Aikawa at Doshisha University, Japan. The pGEX-EEA1$_{36-91}$ clone was obtained from Dr. Bonsu Ku at the Korea Research Institute of Bioscience and Biotechnology (KRIBB). cDNA clones of Rabaptin5 and Rabex-5 were purchased from Open Biosystems. The Rab7 gene was chemically synthesized (Bioneer). Each gene was amplified and cloned into bacterial expression vectors (parallel-His, parallel-GST (*Sheffield et al., 1999*), and pQE30) or mammalian expression vectors (pCDNA-FLAG and pCDNA-HA). Human embryonic kidney 293T (HEK 293T) and HeLa cells were purchased from American Type Culture Collection (RRID: CVCL_0063 and CVCL_0030, respectively) and maintained in Dulbecco's modified Eagle's medium (DMEM) supplemented with 10% fetal bovine serum (heat-inactivated). Plasmids were transfected into cells using Lipofectamine 2000 (Invitrogen) for HeLa cells and PEI 25000 (23966, Polysciences) for HEK 293T cells. Mycoplasma detection kit (LT07-318, Lonza, Switzerland) was regularly used to check absence of mycoplasma.

### Protein expression and purification

GST-ubiquitin G76C, His-Rab5 (WT and mutants), and His-Rabex-5$_{132-393}$ were expressed in *E. coli* BL21(DE3) cells. GST-Rabaptin5$_{551-862}$ and GST-EEA1$_{36-91}$ were expressed in *E. coli* Rosetta2 (DE3) cells. Cells were inoculated and grown to an OD$_{600}$ of 0.6–0.8 at 37 °C. To induce protein expression, 0.5 mM isopropyl D-thiogalactoside (IPTG) was added to the cultures, and they were incubated at 20 °C overnight. Induced cells were harvested by centrifugation at 4,000 rpm. Cells were resuspended in buffer A (50 mM Tris-HCl pH 7.5 and 150 mM NaCl). For GST-Ub G76C and His-Rab5 K-to-C mutants, 2 mM TCEP was added during the purification procedure. Resuspended cells were lysed by sonication and centrifuged at 13,000 rpm. The supernatant was applied to glutathione-*S*-sepharose resin (GE HealthCare) for GST fusion proteins and Ni-NTA-agarose resin (Qiagen) for His-tagged proteins. Non-specific proteins were washed out with buffer B (50 mM Tris-HCl pH 7.5 and 500 mM NaCl) for GST fusion proteins and buffer C (50 mM Tris-HCl pH 7.5, 500 M NaCl, and 20 mM imidazole) for His-tagged proteins. To elute target proteins, buffer D (50 mM Tris-HCl pH 8.0 and 50 mM NaCl, supplemented with 15 mM reduced glutathione) was used for GST-fusion proteins, and buffer E (50 mM Tris-HCl pH 7.5, 300 mM imidazole, and 500 mM NaCl) was used for His-tagged proteins. GST and the His-tag were cleaved from the fusion proteins using GFP-tagged tobacco etch virus protease in cleavage buffer (25 mM Tris-HCl pH 7.5, 75 mM NaCl, and 0.5 mM EDTA) (*Wu et al., 2009*). Cleaved samples were applied to glutathione-*S*-sepharose or Ni-NTA-agarose to remove the released His- or GST- tags. Proteins were further purified by size exclusion chromatography on a Superdex 75 or 200 prep grade 16/60 column (GE HealthCare). Fractions were analyzed by SDS-PAGE, and samples were pooled and concentrated by centrifugal concentrators (Amicon Ultra 3, 10, or 30 kDa, Millipore). Final concentration was determined by either absorbance at 280 nm or Bradford assay.

### Ubiquitination assay

Ubiquitinated Rab5 was detected based on previously described methods (*Choo and Zhang, 2009*; *Jung et al., 2013*). HA-ubiquitin and FLAG-Rab5 were co-transfected into HEK 293T cells. After 24 hr, cells were washed with ice-cold phosphate-buffered saline (PBS) twice and lysed with lysis buffer LA (2% SDS, 150 mM NaCl, 10 mM Tris-HCl, pH 8.0, 2 mM sodium ortho-vanadate, 50 mM sodium fluoride, and a protease inhibitor cocktail from Roche). Lysed cells were transferred into a tube and

boiled for 10 min at 95 °C. The boiled sample was then sonicated (microtip 40% amplitude, pulse 1 s on/1 s off, 20 s, VC 750) and diluted with dilution buffer (150 mM NaCl, 10 mM Tris-HCl, pH 8.0, 2 mM EDTA, and 1% Triton X-100). After centrifugation at 16,200 $g$ for 10 min, the resulting supernatant was pre-cleared with mouse IgG and protein G-beads (GeneDepot). Pre-cleared lysates were immuno-precipitated with FLAG antibody (F1804, Sigma Aldrich, RRID:AB_262044) and immunoblotted with HA-HRP (3F10, Roche, RRID:AB_2314622) or FLAG-IgG-rabbit (F7425, Sigma Aldrich, RRID:AB_439687) followed by Rabbit-IgG-HRP (ADI-SAB-300-J, Enzo Life Sciences, RRID:AB_11179983). For the geranylgeranyl transferase inhibitor assay, 20 μM GGTI-298 (G5169, Sigma) was added for 6 hr after transfection. Twenty-four hours after GGTI treatment, cells were harvested and subjected to ubiquitination assay. These experiments were repeated three times.

## Cytosol/membrane fractionation assay

Cytosol/membrane fractionation assay was performed according to a previously described method (*Baghirova et al., 2015*). Briefly, HEK 293T cells were harvested by trypsin treatment and washed with ice-cold PBS. Fractionation buffer FA (150 mM NaCl, 50 mM HEPES pH 7.4, 100 μg/ml digitonin (D141, Sigma), and 1 M hexylene glycol) supplemented with 1x protease inhibitor cocktail (P8340, Sigma, 100x stock solution) was added to the harvested cells to release cytosolic proteins. After incubation for 15 min on an end-to-end rotator at 4 °C, samples were centrifuged at 2,000 $g$ for 10 min. The resulting supernatant was transferred and marked as cytosolic proteins. Fractionation buffer FB (150 mM NaCl, 50 mM HEPES pH 7.4, 1% (v/v) Nonidet-P40, and 1 M hexylene glycol) supplemented with 1x protease inhibitor cocktail (P8430, Sigma, 100x stock solution) was added to the resulting cell pellets to release membrane proteins. After incubation for 30 min on ice, samples were centrifuged at 7,000 $g$ for 10 min. The resulting supernatant was transferred and marked as membrane proteins. Both cytosolic and membrane fractions were further immunoprecipitated using a FLAG-M2-affinity gel (A2220, Sigma, RRID:AB_10063035) and analyzed by immunoblotting. These experiments were repeated twice.

## EGF-induced EGFR degradation assay

EGFR degradation was assessed based on previously described methods (*Hoeller et al., 2006*; *Balaji et al., 2012*; *Smith et al., 2013*). Briefly, HeLa cells were transfected with FLAG-Rab5 WT, K116R, K140R, or K165R with HA-ubiquitin. After 24 hr of transfection, cells were treated with EGF (20 ng/ml, final concentration) and harvested at the indicated time points after treatment. Harvested cells were lysed with lysis buffer LB (50 mM Tris-HCl pH 7.5, 1% (v/v) NP-40, 150 mM NaCl, and 10% (v/v) glycerol) supplemented with a protease inhibitor cocktail (05-056-489-001, Roche) and subjected to immunoblotting with anti-EGFR antibody (SC-03-rabbit, Santa Cruz, RRID:AB_631420). These experiments were repeated three times.

## Chemical ubiquitination of Rab5

Chemical ubiquitination of Rab5 by formation of a disulfide bond between ubiquitin and a specific lysine residue of Rab5 was based on previously published studies (*Baker et al., 2013a*; *Baker et al., 2013b*; *Merkley et al., 2005*). Purified ubiquitin G76C and a Rab5 K-to-C mutant (K116C, K140C, or K165C) were stored in reducing buffer (2 mM TCEP, 50 mM Tris-HCl pH 7.5, and 150 mM NaCl). The proteins were then mixed at a 1:5 molar ratio (Rab5: ubiquitin) and dialyzed with oxidizing buffer (50 mM Tris-HCl pH 7.5, 150 mM NaCl, 5 mM MgCl$_2$, and 20 μM CuCl$_2$). During dialysis, ubiquitin G76C was added to the dialysis bag at regular intervals. Iterative additions of ubiquitin resulted in 95% yield of monoubiquitinated Rab5 (mUbRab5). Production of mUbRab5 was monitored by SDS-PAGE under non-reducing conditions. Finally, mUbRab5 was purified by size-exclusion chromatography on a Superdex 75 16/60 prep grade column (GE HealthCare) pre-equilibrated with buffer F (50 mM Tris-HCl pH 7.5, 150 mM NaCl, and 10 mM MgCl$_2$).

## Small angle X-ray scattering (SAXS) measurement and data processing

Protein samples were concentrated in Amicon ultra centrifugal devices (3 kDa cut-off). Final concentrations were measured by absorbance at 280 nm. Buffer F was used to record the reference buffer scattering profile. Each sample was measured six times and monitored for radiation damage. Samples were also diluted (two-, four-, and eight-fold) with buffer F to check for concentration

dependency. Scattering profile was generated using in-house software of the Pohang Accelerator Laboratory. $R_g$ (radius of gyration) and $I(0)$ from the Guinier plot were calculated using *AUTORG* (*Svergun, 1992*). *GNOM* was used to calculate the pair distribution function and Porod volume (*Svergun, 1992*). Molecular ensembles were generated by *EOM* (*Tria et al., 2015*). Final ensemble models were cross-validated with the *FoXS* server minimal ensemble search *MES* (*Pelikan et al., 2009*). Since *EOM* accepted only single polypeptide chain with multiple domains, we generated the sequences and atomic coordinates of the mUbRab5s in the following four steps. Firstly, the input sequence started with the ubiquitin sequence followed by the partial Rab5 sequence which covered from the target lysine of Rab5 to the C-terminus and then the remaining Rab5 sequence ranging from the N-terminus to the residue before the target lysine of Rab5. For example, the input sequence for mUbRab5$_{K140}$ was as follows: (N-[Ubiquitin residues 1–76]-[K140-C-term of Rab5]-[N-term to residue 139 of Rab5]-C). Using this sequence, we were able to use *EOM* to analyze the ubiquitin moiety of mUbRab5 as a domain of a single polypeptide chain. Secondly, the input coordinates were prepared following the same way as in the preparation of the input sequences. Crystal structures of ubiquitin and Rab5 were used as the coordinates for domains 1 and 2, respectively (PDB ID: 1UBQ, 3MJH). The residues of Rab5 were re-numbered so that the new numbering was the same as the numbering of the prepared sequence. For example, K140 of Rab5 was re-numbered K77. Thirdly, we determined how many residues in the C-terminal region of ubiquitin should be allowed to be flexible. To define flexible residues in ubiquitin, we analyzed crystal/NMR structures of ubiquitin (PDB IDs: 1UBQ and 1D3Z). In both cases, the C-terminal five residues of ubiquitin (residues 72–76) after the β3 strand were not assigned to secondary structures, which imply that those residues are flexible. Since Arg 72 of ubiquitin is juxtaposed at the end of the β3 strand, Arg 72 was retained in the input coordinate file. Subsequently, four residues of ubiquitin (residues 73–76) were allowed to be flexible and removed in the input coordinate file. These C-terminal four residues of ubiquitin were used for flexibility analysis through *EOM* in our previous study on linear ubiquitin chains (*Thach et al., 2016*). Finally, we used the native isopeptide linkage for ensemble model calculations in *EOM* since a disulfide bond linkage is known to resemble the native ubiquitination linkage (*Baker et al., 2013a*).

## GEF-mediated GDP release and GTP loading assays

GDP release from Rab5 or mUbRab5 was measured using 2'-(or-3')-O-(N-methylanthraniloyl) (MANT)-GDP (Molecular Probes) (*Delprato and Lambright, 2007*; *Baker et al., 2013a*). One micromolar Rab5 or mUbRab5 was loaded with 1 μM MANT-GDP for 30 min at room temperature in buffer F in the presence or absence of 1 μM Rabex-5$_{132-391}$. To initiate MANT-GDP release, 1 mM GDP was added to the solution. Fluorescence intensities were measured (excitation at 360 nm and emission at 450 nm) on a TECAN M200 multi-plate reader, analyzed, and fitted to a one-phase exponential decay curve with *GraphPad Prism* software. GTP loading was measured using MANT-GTP. Two micromolar Rab5 or mUbRab5 was mixed with 1 μM of MANT-GTP in buffer F and stabilized for 10 min. To facilitate GTP loading by GEF, 1 μM Rabex-5$_{132-391}$ was added to the solution. Fluorescence intensities were measured (excitation at 360 nm and emission at 450 nm) on a TECAN M200 multi-plate reader, analyzed, and fitted to a one-phase exponential association curve using *GraphPad Prism* software. These experiments were repeated three times.

## Effector protein binding assay by biolayer interferometry (BLI)

BLI experiments were performed on a BLitz system (ForteBio). For this, 0.5 μM GST-Rabaptin5$_{551-862}$ or GST-EEA1$_{36-91}$ was immobilized on anti-GST sensors. After baseline stabilization with reaction buffer (50 mM Tris-HCl, 150 mM NaCl, 10 mM MgCl$_2$, and 1 mM 5'-guanylyl imidodiphosphate, a non-hydrolysable GTP analog (GppNHP, Jena Bioscience, Germany)), GppNHP-loaded Rab5 or mUbRab5 at different concentrations was loaded until signal saturation, and this experiment was repeated twice. To initiate dissociation, the sensor was then placed into in reaction buffer again. Association rate constant ($k_{on}$), dissociation rate constant ($k_{off}$), and SEM (standard error of mean) values were calculated by *BLItz Pro* (ForteBio) based on sensorgram results. Dissociation constants ($K_d$) and SEMs of dissociation constants were calculated by the following equation:

$$K_d = K_{off}/K_{on},$$

$$\Delta K_d = K_d \sqrt{\left(\frac{\Delta k_{on}}{k_{on}}\right)^2 + \left(\frac{\Delta k_{off}}{k_{off}}\right)^2}$$

where $\Delta$ refers to the SEMs of $K_d$, $k_{on}$, and $k_{off}$, respectively.

## Immunofluorescence

After transfection, cells were rinsed in PBS and fixed for 10 min at room temperature with 3% para-formaldehyde. Cells were then permeabilized with 0.1% Triton X-100 in PBS for 5 min and blocked with 2% bovine serum in PBS for 1 hr at room temperature. For immunostaining, cells were incubated with antibodies diluted in 2% bovine serum at room temperature for 1 hr, then rinsed with PBS and incubated with either anti-rabbit-Rhodamine or anti-mouse-Alexa-Fluor-633-conjugated secondary antibodies (R6394;RRID:AB_1500693, A11001;RRID:AB_2534069, respectively, Life Technologies,) for 30 min at room temperature. Cells were washed three times for 5 min with PBS. Images were obtained using a LSM-700 (Zeiss).

## Acknowledgements

We thank the staff members at beamline 4C of Pohang Accelerator Laboratory for technical assistance in SAXS data collection. This work was supported by the Basic Science Research Program (NRF-2015R1A2A1A15055951), the Science Research Center Program (SRC-2017R1A5A1014560), and the Pioneer Research Center Program (2012–0009597) through National Research Foundation of Korea (NRF) grants funded by the Korea Ministry of Science and ICT and by the Woo Jang Chun program (PJ009106) through the Rural Development Agency.

## Additional information

### Funding

| Funder | Grant reference number | Author |
|---|---|---|
| National Research Foundation of Korea | NRF-2015R1A2A1A15055951 | Donghyuk Shin<br>Gyuhee Kim<br>Jiseok Baek<br>Sangho Lee |
| Rural Development Administration | PJ009106 | Donghyuk Shin<br>Gyuhee Kim<br>Jiseok Baek<br>Sangho Lee |
| National Research Foundation of Korea | 2012-0009597 | Donghyuk Shin<br>Gyuhee Kim<br>Jiseok Baek<br>Sangho Lee |
| National Research Foundation of Korea | SRC-2017R1A5A1014560 | Donghyuk Shin<br>Wooju Na<br>Ji-Hyung Lee<br>Gyuhee Kim<br>Jiseok Baek<br>Seok Hee Park<br>Cheol Yong Choi<br>Sangho Lee |

The funders had no role in study design, data collection and interpretation, or the decision to submit the work for publication.

### Author contributions

Donghyuk Shin, Conceptualization, Investigation, Methodology, Writing—original draft, Writing—review and editing; Wooju Na, Ji-Hyung Lee, Jiseok Baek, Investigation, Methodology; Gyuhee Kim, Resources, Methodology; Seok Hee Park, Cheol Yong Choi, Data curation, Investigation; Sangho

Lee, Conceptualization, Supervision, Funding acquisition, Writing—original draft, Project administration, Writing—review and editing

## Author ORCIDs

Donghyuk Shin ⓘ http://orcid.org/0000-0002-8272-6133
Sangho Lee ⓘ http://orcid.org/0000-0003-3886-4579

## Decision letter and Author response

Decision letter https://doi.org/10.7554/eLife.29154.028
Author response https://doi.org/10.7554/eLife.29154.029

## Additional files

### Supplementary files

• Transparent reporting form
DOI: https://doi.org/10.7554/eLife.29154.027

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
