## [Decision Letter]

Thank you for submitting your article "Site–specific monoubiquitination downregulates Rab5 by disturbing effector binding and guanine nucleotide conversion" for consideration by *eLife*. Your article has been reviewed by three peer reviewers, one of whom is a member of our Board of Reviewing Editors and the evaluation has been overseen by Ivan Dikic as the Senior Editor. The following individuals involved in review of your submission have agreed to reveal their identity: Marino Zerial (Reviewer #2); David Lambright (Reviewer #3).

You will see that the reviewers believe that the work is potentially suitable for presentation in *eLife*. However, they describe a number of aspects that need to be fully addressed and they noted that the manuscript needs rewriting, perhaps with help from a native English speaker. I enclose the full reviews to help in recrafting this story for submission and re–evaluation for presentation in *eLife*. We hope you find the comments constructive; note that the ubiquitination of Rab5 is already reported in PhosphoSite Plus mass spec database (when you respond to reviewer 3); the Rab7 story can be omitted.

Reviewer #1

This is an interesting paper reporting the consequences of Rab5 and Rab7 monoubiquitynation on Rab protein function. The authors try to combine cell biological assays and biochemical analysis with SAXS structural analysis, the kind of broad mix of approaches that *eLife* would like to present. They present a method for efficient in vitro generation of pure mono–ubiquitin conjugates. It is likely that the authors have shown that mono–ubiquitination decreases Rab activity when driven in cells by artificial expression of extra HA–ubiquitin. This is shown with direct binding assays for Rab5 effectors and some light microscopy.

Unfortunately, it appears that the figures were photographs of drawings and are of too poor resolution for publication and in some cases even for reviewer evaluation. This reviewer cannot see any EEA1 staining for example, as the images are out of focus or off too low a resolution to see even blown up on a computer screen. Also, the language is difficult to understand and the manuscript will need significant editing and rewriting for clarity. The EGFR degradation needs to be presented graphically so that half times for degradation relative to time zero can be compared.

I will await the other referee comments and hope that this story can be presented more clearly so that its findings can be appreciated by the scientific community and brought up to the standard required for presentation in *eLife*.

(Please use photoshop, high resolution and a drawing program to label figures such as Adobe Illustrator)

Quantitation of the cell biology data is missing

Reviewer #2

In this study, the authors employ a range of techniques to investigate Rab5 monoubiquitination both functionally and structurally, culminating in a model of how differential monoubiquitination of Rab5 has different effects on the downstream Rab5 machinery and dynamics at the early endosome. While the premise of this manuscript is highly interesting and is in the scope of *eLife*, I do not believe that the manuscript in the current form meets the standards of this journal. With sufficient revisions, however, this could be a conceptually novel and compelling paper.

In general, the story presented here is not very coherent. The authors have performed all the right experiments to present a good case for their model. However, the experiments are often poorly described and presented in an order that is only logical if you already know the outcome of later experiments. The authors should take care to present their story in a stepwise and logical manner and to explain in detail both the experimental approaches and underlying logic.

General points:

1) It is essential to be clear about which cell lines are used and also provide the rationale for using two different cell lines for the biochemistry and microscopy experiments. This approach is not coherent. Similarly, the change from cell work to recombinant proteins is not made explicit in the text.

2) The authors should go through the manuscript carefully to fix grammatical mistakes. Especially in the case of subsection “Monoubiquitination on K140 of Rab5 decreases its binding affinity to effector prot”, grammatical errors may confuse the message the authors are trying to convey.

There are several Rab5 isoforms and it must be made clear from the start which one is being used and provide a sound rationale for using that specific isoform alone, as the three isoforms show differential ubiquitination patterns.

Next, the authors use GGTI–198, a GGTase inhibitor, to rule out prenylation as the source of the double bands upon ubiquitination. However, this experiment is not sufficiently described in the figure legend or in the text. What are the two different IP FLAG plots in Figure 1, what do the labels short and long mean?

Importantly, it is not clear to the reader why out of all lysines in Rab5, K33 and K165 were chosen in addition to the known K116, K134 and K140 sites from prot*EOM*ics. There is some explanation for K165 in sequence homology to K147 in Ras, but otherwise the authors make no effort to explain their rationale. This section should be reordered to make it more logical. I would suggest reordering the first part of the manuscript: the story would flow better if the validation of prot*EOM*ics hits preceded the investigation of the now proven ubiquitination sites. For example, instead of starting with candidate ubiquitination sites and an analysis of crystal structures in subsection “Rab5 undergoes monoubiquitination at multiple sites”, it would be more logical to begin the experiments presented in Figure 1 as a validation of published prot*EOM*ics data. From there the story can then focus on the relevant monoubiquitination sites and their location on the Rab5 structure (Figure 1—figure supplement 1).

Furthermore, the experiments done with K to E mutations and protein expression are not essential and should be moved to the supplement or left out entirely so as not to distract from the more sophisticated K to R mutation experiments.

In Figure 2, the authors show that monoubiquitinated Rab5 (mUbRab5) localises to membranes. This raises many questions: mUbRab5 is a very small percentage of total membrane Rab5, why do the authors observe such a dramatic effect on recycling/degradation? This would seem to be a highly significant result but is not addressed at all. In addition to this, Figure 2 shows a dramatic removal of Rab5 from EEA1 positive endosomes upon ubiquitination and indeed in the text the authors refer to this as 'diffusing to cytosol' which would not agree with the results presented in Figure 2.

If K134 mutations are not viable, how has this been identified as a ubiquitination site in vivo? The authors should comment on this.

The authors introduce the immunofluorescence of Rab5 endosomes and justify the use of HeLa cells in subsection “Monoubiquitination on K140 and K165 of Rab5 plays negative roles in endocytic pathway”, but results from these experiments had already been shown in subsection “Rab5 undergoes monoubiquitination at multiple sites” without explanation of what Rab5 puncta are or what cell line was used.

In Figure 2, UbRab5 does not appear as a double band as in other blots, why is this?

Subsection “Monoubiquitination on K140 and K165 of Rab5 plays negative roles in endocytic pathway” is a bit of a reach of interpretation at this point in the manuscript and are only logical in hindsight, knowing the results that are still to come.

From the SAXS analysis the authors derive that mUbRab5_K140_ features two conformations while mUbRab5_K116_ and mUbRab5_K165_ have only one conformation. However, the authors also state in the text that mUbRab5 is structurally very flexible, contradicting the existence of distinct conformations. To better evaluate the data, the authors need to explain in detail how distinct conformations were extracted from the data using ensemble–optimized methods (*EOM*) and how the final models were derived (Figure 4). Furthermore, the authors need to show the shape of the particle derived from the SAXS data, not only the model. In general, the unprocessed SAXS data are missing and the reader is presented immediately with something highly processed. For a more coherent story, the intermediate stages of deriving these structural models must be presented.

From structural models the authors derive that the close proximity of the ubiquitin moieties in mUbRab5_K140_ to the switch I region of Rab5 might disturb Rabaptin–5 and EEA1 binding, while interaction of Rab5 with GEF (Rabex–5) and GAP (RabGAP–5) is unchanged. This especially needs clarification, because the switch I region is involved in nucleotide binding as well, and hence prone to be influenced by changes in nucleotide state. Knowing the outcome of the binding experiments described later, this conclusion makes sense, but when reading this manuscript for this first time this appears a somewhat illogical conclusion.

Reviewer #3

This manuscript describes experiments examining monoubiquitination of Rab5 when overexpressed with ubiquitin and affects of monoubiquitination of the specific–lysine residues on Rab5 structure, localization, EGFR degradation, intrinsic and Rabex–5 catalyzed nucleotide exchange, and binding of effectors (Rabaptin5 and EEA1). Similar cell–based experiments were performed to examine Rab7 ubiquitination. The identification of putative monoubiquitination sites, the characterization of homogenously monoubiquitinated Rab5 by SAXS, and the effects of monoubiquitination on GEF activity, effector binding affinity, localization, and EGFR degradation suggest that monoubiquitination of at least two lysines could potentially roles regulate Rab5 and Rab7 functions.

All of the experiments involve either purified proteins or transfections of cell lines. There are no experiments assessing whether the endogenous Rab proteins are monoubiquinated (i.e. without ubiquitin overexpression) and under what conditions. This limitation should be discussed, and conclusions about the potential biological significance appropriately qualified.

Subsection “Rab5 undergoes monoubiquitination at multiple sites”, K33 and K134 are indeed crucial but this is already known, indeed from the very first structures of EF–Tu and Ras. K33 is the P–loop lysine in the GxxxxGKS/T motif; K134 is the invariant lysine in the NKxD G4 motif. The role and interactions of these residues in GTPases in general has been extensively documented. Its not clear why these invariant residues were chosen for mutational analysis. Expression in inclusion bodies in bacteria isn't particularly informative or unexpected. The rationale for generating lysine–to–glutamate mutations is also unclear as these mutations don't mimic ubiquitination.

How can anything be concluded about inhibition of Rab5 activation (subsection “Monoubiquitination on K140 and K165 of Rab5 plays negative roles in endocytic pathway”) from the IF and fractionation experiments in Figure 2? These experiments do not indicate whether ubiquitination affects the nucleotide status of Rab5. Perhaps the authors meant inhibition of a Rab5 function rather than activation of Rab5 per se. Needs clarification.

Overexpression of ubiquitin presumably affects the function of many targets including proteins other than Rab5 that are also important for endocytic trafficking. Does overexpression of ubiquitin alone have any effects on (for example) EGFR degration?

The cell-based experiments lack quantification and analysis of statistical significance.

The resolution of the images in the PDF is low, making it difficult to evaluate the quality of these experiments.

The buffer–subtracted SAXS curves should be shown somewhere in the main or supplemental figures. Accurate modeling of SAXS data, in particular by ensemble approaches, requires data with high signal to noise that typically extends to a high resolution cutoff near the first Porod minimum. The data and calculated curves in Figure 4 are truncated in the Kratky region and the signal–to–noise does not appear to be very high. Are the data of sufficient quality for ensemble modeling?

No details are provided regarding the *EOM* modeling. For example, how many residues we're allowed to be flexible and what criteria where to determine which residues show be flexible? Do the results change if additional residues are allowed to be flexible? A narrow selected distribution near the high *R_g_* and *D_max_* tails of the pools in Figure 4 suggest that the number of flexible residues used to generate the pool might not be adequate to represent the flexibility in solution. Also unclear is how the linkage between Rab5 and ubiquitin is modeled. Does the *EOM* model disulfide linkages? If not, how was the linkage modeled? The details of the modeling are important and how they affect the results needs to be evaluated and discussed.

"Interestingly, none of the *EOM*–derived structural models for mUbRab5s implicates any direct interaction between ubiquitin and Rab5" (subsection “Structural models of mUbRab5s derived by solution scattering”). Is *EOM* capable of producing models with direct interactions? Even if it could, what would it mean at the low resolution of SAXS?

Figure 4—figure supplement 1: Kratky–Debye and Porod–Debye plots indicate how extended a molecule is. Flexibility is inferred. Presumably the same results would be obtained from the calculated *EOM* curves that fit the data. The question is whether the SAXS data can distinguish flexible vs. extended. The interpretation should be more nuanced.

---

## [Author Response]

Reviewer #1[…] Unfortunately, it appears that the figures were photographs of drawings and are of too poor resolution for publication and in some cases even for reviewer evaluation.

We are submitting our revised manuscript with high resolution image files.

This reviewer cannot see any EEA1 staining for example, as the images are out of focus or off too low a resolution to see even blown up on a computer screen. Also, the language is difficult to understand and the manuscript will need significant editing and rewriting for clarity.

The revised manuscript was re–written and has been edited by native English speaker. We would be delighted to provide you a certificate by the editing company upon request.

The EGFR degradation needs to be presented graphically so that half times for degradation relative to time zero can be compared.

The graphical representation of EGFR degradation has been included in the revised manuscript (Figure 2).

I will await the other referee comments and hope that this story can be presented more clearly so that its findings can be appreciated by the scientific community and brought up to the standard required for presentation in eLife.(Please use photoshop, high resolution and a drawing program to label figures such as Adobe Illustrator)

We are now providing high resolution images prepared by Adobe Photoshop and Illustrator as the reviewer requested.

Quantitation of the cell biology data is missing

We have added statistics and the significance for cell-based experiments in our revised manuscript (Figure 2).

*Reviewer #2*

[…] 1) It is essential to be clear about which cell lines are used and also provide the rationale for using two different cell lines for the biochemistry and microscopy experiments. This approach is not coherent. Similarly, the change from cell work to recombinant proteins is not made explicit in the text.

The rationale for the transition from HEK293T cells to HeLa cells is described in the revised manuscript as follows:

(Subsection “Rab5 undergoes monoubiquitination at multiple sites”) “Subsequently, we generated an 11KR mutant with intact K33 and K134 and examined its function by observing Rab5–positive endosome puncta in HeLa cells to visualize Rab5–positive endosomes more efficiently (Miaczynska et al., 2004, Borg et al., 2014, Kajiho et al., 2003).”

The rationale for the transition from cell work to recombinant proteins is described in the revised manuscript as follows:

(Subsection “Modified chemical ubiquitination with iterative ubiquitin addition to obtain fully monoubiquitinated Rab5 protein”) “To understand how monoubiquitinated Rab5s negatively affect the Rab5 regulatory cycle at molecular level, we undertook biochemical studies of monoubiquitinated Rab5s.”

2) The authors should go through the manuscript carefully to fix grammatical mistakes. Especially in the case of subsection “Monoubiquitination on K140 of Rab5 decreases its binding affinity to effector prot”, grammatical errors may confuse the message the authors are trying to convey.

We thank the reviewer for careful reading of the manuscript. The revised manuscript has been edited by a native English speaker through a professional editing service company.

There are several Rab5 isoforms and it must be made clear from the start which one is being used and provide a sound rationale for using that specific isoform alone, as the three isoforms show differential ubiquitination patterns.

We used Rab5a in this study. Since Rab5a is extensively studied compared to Rab5b and Rab5c, we decided to use the Rab5a isoform. We revised the manuscript as follows:

(Introduction) “Three isoforms of mammalian Rab5 – Rab5a, Rab5b and Rab5c regulate endocytosis in a co–operative manner (Bucci et al., 1995). However, recent studies suggest that these three isoforms may have differential roles. Rab5a is required for EGF–induced EGFR degradation while Rab5b and Rab5c are not (Barbieri et al., 2000a, Chen et al., 2009). Rab5c is involved in cell migration whereas Rab5a and Rab5b are not (Chen, Schauer, et al., 2014). Prot*EOM*ic studies have suggested that the Rab5 isoforms can be differentially ubiquitinated in cells (Chen, Zhou, et al., 2014, Wagner et al., 2011, Wagner et al., 2012). Since Rab5a has been extensively studied for its involvement in endosomal fusion (Bucci et al., 1992, Gorvel et al., 1991, Barbieri et al., 2000b, Hoffenberg et al., 1995, Rybin et al., 1996, Stenmark et al., 1994), we decided to focus on Rab5a for our studies (hereafter referred to as ‘Rab5’). Here, we identified monoubiquitination sites of Rab5 in cultured cells.”

Next, the authors use GGTI–198, a GGTase inhibitor, to rule out prenylation as the source of the double bands upon ubiquitination. However, this experiment is not sufficiently described in the figure legend or in the text. What are the two different IP FLAG plots in Figure 1, what do the labels short and long mean?

The treatment of GGTI–198 is described in the Materials and methods section as follows:

(Subsection “Ubiquitination assay”) “For the geranylgeranyl transferase inhibitor assay, 20 μM GGTI–298 (G5169, Sigma) was added for 6 hours after transfection. Twenty–four hours after GGTI treatment, cells were harvested and subjected to ubiquitination assay. These experiments were repeated three times.”

We described that label in Figure 1 legend as follows:

(Figure 1 legend) “Short and long refer to relative exposure time during immunoblotting.”

Importantly, it is not clear to the reader why out of all lysines in Rab5, K33 and K165 were chosen in addition to the known K116, K134 and K140 sites from protEOMics. There is some explanation for K165 in sequence homology to K147 in Ras, but otherwise the authors make no effort to explain their rationale. This section should be reordered to make it more logical. I would suggest reordering the first part of the manuscript: the story would flow better if the validation of protEOMics hits preceded the investigation of the now proven ubiquitination sites. For example, instead of starting with candidate ubiquitination sites and an analysis of crystal structures in subsection “Rab5 undergoes monoubiquitination at multiple sites”, it would be more logical to begin the experiments presented in Figure 1 as a validation of published protEOMics data. From there the story can then focus on the relevant monoubiquitination sites and their location on the Rab5 structure (Figure 1—figure supplement 1).

We completely revised the manuscript according to the logical flow the reviewer suggested. Briefly, ubiquitination sites from prot*EOM*ics studies (K116, K134 and K140) were validated with ubiquitination assay. We found that K116 and K140 were monoubiquitinated. By contrast, K134 was not responsible for monoubiquitination on Rab5, implicating that K134 might undergo polyubiquitination. Another monoubiquitination site K165, homologous to K147 of Ras, was identified. Finally the three putative monoubiquitination sites (K116, K140 and K165) were chosen for further studies. These changes have been implemented in the lines 78–152 of the revised manuscript.

Furthermore, the experiments done with K to E mutations and protein expression are not essential and should be moved to the supplement or left out entirely so as not to distract from the more sophisticated K to R mutation experiments.

Pursuant to the reviewer’s suggestion, the results on K to E mutation and their expression are now omitted from the revised manuscript.

In Figure 2, the authors show that monoubiquitinated Rab5 (mUbRab5) localises to membranes. This raises many questions: mUbRab5 is a very small percentage of total membrane Rab5, why do the authors observe such a dramatic effect on recycling/degradation? This would seem to be a highly significant result but is not addressed at all. In addition to this, Figure 2 shows a dramatic removal of Rab5 from EEA1 positive endosomes upon ubiquitination and indeed in the text the authors refer to this as 'diffusing to cytosol' which would not agree with the results presented in Figure 2.

The dramatic effects of a small percentage of monoubiquitinated Rab5 on recycling/degradation is now discussed in the revised manuscript as follows:

(Discussion) “In this study, we found that monoubiquitination of Rab5 had an inhibitory effect on the regulatory cycle of Rab5. However, we did not find a stimulus signal or E3 ligase(s) responsible for monoubiquitination of Rab5. As result, only a small portion of Rab5 could be ubiquitinated when ubiquitin was overexpressed. These results suggest that ubiquitination of Rab5 occurs rarely in cells under basal conditions and might the reason why the ubiquitination of Rab GTPases has not been studied until recently (de la Vega et al., 2011). Despite the small population of monoubiquitinated Rab5, the effects of monoubiquitination of Rab5 were strong. We observed dramatic failure of localization of Rab5 to the endosome and EGFR degradation pathway. This suggests that monoubiquitination of Rab5 is tightly regulated in the cell through specific E3 ligase or DUB enzymes. It will be of great interest to determine how monoubiquitination of Rab5 is regulated in cells.”

The confusions for describing localization of Rab5 upon ubiquitin transfection is now clarified as follows:

(Subsection “Monoubiquitination of K140 and K165 of Rab5 plays a negative role in the endocytic pathway”) “We observed that the monoubiquitinated Rab5 was predominantly localized in the membrane fraction. These results prompted us to hypothesize that monoubiquitinated Rab5 could disrupt Rab5–positive endosomal formation while it was localized on the membrane. Together with the immunofluorescence assay results (Figure 2), these findings strongly suggest that ubiquitination of Rab5 has negative effects on the regulatory cycle of Rab5.”

If K134 mutations are not viable, how has this been identified as a ubiquitination site in vivo? The authors should comment on this.

Our ubiquitination assay data suggests that K134 is not monoubiquitinated. Since K134 was previously reported as an ubiquitination site by prot*EOM*ic studies, it is likely that K134 may be polyubiquitinated, although we have no direct evidence for it. Polyubiquitination of K134 would destabilize the structural integrity of Rab5, ultimately leading to its degradation. We have commented about this concern as follows:

(Subsection “Rab5 undergoes monoubiquitination at multiple sites”) “No monoubiquitinated Rab5 band was detected for the 11KR mutant in the ubiquitination assay (Figure 1), suggesting that K33 and K134 are not responsible for monoubiquitination of Rab5. Considering that K134 was previously reported to be ubiquitinated by prot*EOM*ic studies (Chen, Zhou, et al., 2014, Wagner et al., 2011, Wagner et al., 2012), it is plausible that K134 may undergo polyubiquitination. Given the crucial role of K134 in the structural integrity of Rab5 (Figure 1—figure supplement 1), we hypothesized that ubiquitination of K134 might be related to disruption/degradation of Rab5, thereby preventing us from observing ubiquitination of K134 in the current experimental conditions.”

The authors introduce the immunofluorescence of Rab5 endosomes and justify the use of HeLa cells in subsection “Monoubiquitination on K140 and K165 of Rab5 plays negative roles in endocytic pathway”, but results from these experiments had already been shown in subsection “Rab5 undergoes monoubiquitination at multiple sites” without explanation of what Rab5 puncta are or what cell line was used.

We revised our manuscript guided by the reviewer’s suggestion as follows:

(Subsection “Rab5 undergoes monoubiquitination at multiple sites”) “Subsequently, we generated an 11KR mutant with intact K33 and K134 and examined its function by observing Rab5–positive endosome puncta in HeLa cells to visualize Rab5–positive endosomes more efficiently (Miaczynska et al., 2004, Borg et al., 2014, Kajiho et al., 2003).”

In Figure 2, UbRab5 does not appear as a double band as in other blots, why is this?

Different percentage of gel was used for Figure 2 (10% gel for Figure 1, 12% gel for Figure 2) since we already confirmed that both double bands are responsible for monoubiquitinated Rab5. This point is explicitly stated in the revised manuscript as follows:

(Subsection “Rab5 undergoes monoubiquitination at multiple sites”) “These double bands were not observed when a high percentage polyacrylamide gel (12%) was used (Figure 1).”

Subsection “Monoubiquitination on K140 and K165 of Rab5 plays negative roles in endocytic pathway” are a bit of a reach of interpretation at this point in the manuscript and are only logical in hindsight, knowing the results that are still to come.

We removed these sentences in the revised manuscript, and the preceding paragraph was entirely re–written as the reviewer suggested.

From the SAXS analysis the authors derive that mUbRab5_K140_ features two conformations while mUbRab5_K116_ and mUbRab5_K165_ have only one conformation. However, the authors also state in the text that mUbRab5 is structurally very flexible, contradicting the existence of distinct conformations. To better evaluate the data, the authors need to explain in detail how distinct conformations were extracted from the data using ensemble–optimized methods (EOM) and how the final models were derived (Figure 4). Furthermore, the authors need to show the shape of the particle derived from the SAXS data, not only the model. In general, the unprocessed SAXS data are missing and the reader is presented immediately with something highly processed. For a more coherent story, the intermediate stages of deriving these structural models must be presented.

To determine whether we can apply ab initio modeling of mUbRab5s, we carefully analyzed our SAXS curves of mUbRab5s (described in subsection “Structural models of mUbRab5s derived by solution scattering”). As shown in Figure 1, all the mUbRab5s feature a loose plateau in the Porod–Debye plot, supporting that all the mUbRab5s are flexible in solution. Because mUbRab5 shows clear feature of flexibility, we decided to analyze the SAXS curve with *EOM*. The details of how we extracted distinct conformations of mUbRab5s through *EOM* have been described as follows:

(Subsection “Structural models of mUbRab5s derived by solution scattering”) “*EOM* was originally designed for multi–domain proteins with a flexible linker. Input files for *EOM* are a single linear protein sequence covering entire protein, atomic coordinates for each domain, and SAXS data (Tria et al., 2015). Because ubiquitinated proteins are not single polypeptides but branched ones, we generated the sequences and atomic coordinates of mUbRab5s as described in the Materials and methods. A pool of 10,000 independent models was generated based on the sequence and structural information from SAXS curve by *RANCH* (embedded in *EOM, ATSAS* package). Then, a genetic algorithm for the selection of an ensemble was performed by *GAJOE* with 100 times (embedded in *EOM, ATSAS* package). Finally, the best ensemble matched with the SAXS curve with lowest χ^2^ was selected."

(Subsection “Small angle X-ray scattering (SAXS) measurement and data processing”) “Since *EOM* accepted only single polypeptide chain with multiple domains, we generated the sequences and atomic coordinates of the mUbRab5s in the following four steps. [...] Finally, we used the native isopeptide linkage for ensemble model calculations in *EOM* since a disulfide bond linkage is known to resemble the native ubiquitination linkage (Baker, Lewis, et al., 2013).”

Based on the *ab initio* model from initial SAXS curve one can obtain an idea for the shape of the particle. Therefore, we have generated ab initio models for mUbRab5s and the results were deposited on SASBDB. (Since the data will be released after acceptance on publication, you can only access them through the link below.)

mUbRab5_K116_: https://www.sasbdb.org/data/SASDCM6/tkmwsdjl7l/

mUbRab5_K140_: https://www.sasbdb.org/data/SASDCN6/0ec25yspdo/

mUbRab5_K165_: https://www.sasbdb.org/data/SASDCP6/7wnrjfzgyv

We question whether we should include the ab initio models in the revised manuscript. Technically ab initio modelling should be performed with monodisperse sample, which is not flexible and does not have oligomeric state. However, mUbRab5s revealed signs of high flexibility (Figure 1). If one generates ab initio model with a flexible sample, the overall size of the molecule will be much larger than expected, and the χ^2^ value with an atomic model will be poorer than the value generated by the *EOM* analysis. In addition, analyzing and showing a SAXS curve with two different systems (in this case, flexible and ab initio) is technically improper. One should choose the best method that describes their SAXS curve. With this regards, we suggest to include our SASBDB deposition ID numbers along with the *EOM* analysis in the revised manuscript.

Since the resolution of our figures was not sufficient in the initially submitted manuscript, it might be hard to find the unprocessed SAXS data which is shown as the black dots in Figure 4 (labeled as “Experiment”). We hope that the high resolution Figure 4 in the revised manuscript will resolve this issue.

From structural models the authors derive that the close proximity of the ubiquitin moieties in mUbRab5_K140_ to the switch I region of Rab5 might disturb Rabaptin–5 and EEA1 binding, while interaction of Rab5 with GEF (Rabex–5) and GAP (RabGAP–5) is unchanged. This especially needs clarification, because the switch I region is involved in nucleotide binding as well, and hence prone to be influenced by changes in nucleotide state. Knowing the outcome of the binding experiments described later, this conclusion makes sense, but when reading this manuscript for this first time this appears a somewhat illogical conclusion.

As the reviewer suggested, we clarified the paragraph with logical flow as below. Briefly, the conformational change of SWI is interrupted by ubiquitin moiety, which would cause the activation of Rab5 through interaction with downstream effector proteins. We revised the manuscript as follows:

(Subsection “Structural models of mUbRab5s derived by solution scattering”) “However, the switch I (SWI) region of Rab5 was juxtaposed with the flexible ubiquitin moiety of mUbRab5_K140_, raising the possibility that a flexible ubiquitin moiety could cause a conformational change in the SWI. Because this conformational change of SWI is required for activation of Rab5 through interactions with downstream effector proteins, potential interruption of the SWI by mUbRab5_K140_ could have a severe impact on the Rab5 regulatory cycle.

Reviewer #3[…]All of the experiments involve either purified proteins or transfections of cell lines. There are no experiments assessing whether the endogenous Rab proteins are monoubiquinated (i.e. without ubiquitin overexpression) and under what conditions. This limitation should be discussed, and conclusions about the potential biological significance appropriately qualified.

We appreciate the reviewer’s careful reading of our manuscript and suggestion to state the limitation of our study. Pursuant to the reviewer’s suggestion, we discussed the limitation that no experiment on the monoubiquitination of endogenous Rab without ubiquitin overexpression and the biological significance in the revised manuscript as follows:

(Discussion) “In this study, we found that monoubiquitination of Rab5 had an inhibitory effect on the regulatory cycle of Rab5. However, we did not find a stimulus signal or E3 ligase(s) responsible for monoubiquitination of Rab5. As result, only a small portion of Rab5 could be ubiquitinated when ubiquitin was overexpressed. These results suggest that ubiquitination of Rab5 occurs rarely in cells under basal conditions and might the reason why the ubiquitination of Rab GTPases has not been studied until recently (de la Vega et al., 2011). Despite the small population of monoubiquitinated Rab5, the effects of monoubiquitination of Rab5 were strong. We observed dramatic failure of localization of Rab5 to the endosome and EGFR degradation pathway. This suggests that monoubiquitination of Rab5 is tightly regulated in the cell through specific E3 ligase or DUB enzymes. It will be of great interest to determine how monoubiquitination of Rab5 is regulated in cells.”

Subsection “Rab5 undergoes monoubiquitination at multiple sites”, K33 and K134 are indeed crucial but this is already known, indeed from the very first structures of EF–Tu and Ras. K33 is the P–loop lysine in the GxxxxGKS/T motif; K134 is the invariant lysine in the NKxD G4 motif. The role and interactions of these residues in GTPases in general has been extensively documented. Its not clear why these invariant residues were chosen for mutational analysis. Expression in inclusion bodies in bacteria isn't particularly informative or unexpected. The rationale for generating lysine–to–glutamate mutations is also unclear as these mutations don't mimic ubiquitination.

We have cited the references as the reviewer suggested and revised our manuscript as below. In addition, the lysine–to–glutamate mutation studies are removed in the revised manuscript.

(Subsection “Rab5 undergoes monoubiquitination at multiple sites”) “However, the 13KR mutant was not expressed in HEK 293T cells (Figure 1). We reasoned that this was due to mutation of both K33 and K134 based on previous structural studies of Ras and EF–Tu GTPases (Pai et al., 1989, Brunger et al., 1990, Jurnak, 1985, Berchtold et al., 1993). Two lysine residues, the P–loop lysine in the GxxxGKS/T motif and the invariant lysine in the NKxD motif, are crucial because they form direct contacts with the guanine nucleotide. Similarly, for Rab5, K33 and K134 form direct contacts with both GDP and GTP molecules (Figure 1—figure supplement 1, PDB ID: 1TU4 and 3MJH, respectively).”

How can anything be concluded about inhibition of Rab5 activation (subsection “Monoubiquitination on K140 and K165 of Rab5 plays negative roles in endocytic pathway”) from the IF and fractionation experiments in Figure 2? These experiments do not indicate whether ubiquitination affects the nucleotide status of Rab5. Perhaps the authors meant inhibition of a Rab5 function rather than activation of Rab5 per se. Needs clarification.

We performed ubiquitination assay with Rab5 GDP–/GTP– bound mutants (S34N and Q79L, respectively) and the results were presented in Figure 1. Since we found clear monoubiquitination band from both S34N and Q79L mutants, we concluded that the monoubiquitination is independent of the nucleotide binding status of Rab5. However, as the reviewer commented, we cannot definitively indicate whether ubiquitination on Rab5 affects the nucleotide status of Rab5. Therefore, we revised the expression from “activation” to “inhibition” of Rab5, and it is described in the revised manuscript as follows:

(Subsection “Monoubiquitination of K140 and K165 of Rab5 plays a negative role in the endocytic pathway”) “Together with the immunofluorescence assay results (Figure 2), these findings strongly suggest that ubiquitination of Rab5 has negative effects on the regulatory cycle of Rab5.”

Overexpression of ubiquitin presumably affects the function of many targets including proteins other than Rab5 that are also important for endocytic trafficking. Does overexpression of ubiquitin alone have any effects on (for example) EGFR degration?

We agree with the reviewer that ubiquitin overexpression could affect other proteins in the endocytic trafficking. To explore such possibility, we examined whether overexpression of ubiquitin alone affects the EGFR degradation (Figure 2). We observed no significant difference in the EGFR degradation by ubiquitin overexpression itself. While we cannot exclude the possibility that ubiquitin overexpression may affect proteins in the endocytic pathways, our results with K–to–R or single K mutants together with biochemical assay strongly support that these observations are tightly linked with ubiquitination of Rab5 (Figure 5, Figure 6).

The cell-based experiments lack quantification and analysis of statistical significance.

We added statistics and the significance for cell-based experiments in our revised manuscript (Figure 2).

The resolution of the images in the PDF is low, making it difficult to evaluate the quality of these experiments.

We are submitting our revised manuscript with high resolution image files.

The buffer–subtracted SAXS curves should be shown somewhere in the main or supplemental figures. Accurate modeling of SAXS data, in particular by ensemble approaches, requires data with high signal to noise that typically extends to a high resolution cutoff near the first Porod minimum. The data and calculated curves in Figure 4 are truncated in the Kratky region and the signal–to–noise does not appear to be very high. Are the data of sufficient quality for ensemble modeling?

Since the resolution of our figures was not sufficient in the initially submitted manuscript, it might be hard to find the unprocessed SAXS data which is shown as the black dots in Figure 4 (labeled as “Experiment”). We hope that the high resolution Figure 4 in the revised manuscript will resolve this issue.

The experimental setup at the synchrotron beamline where we collected SAXS data prevented us from obtaining high resolution data. However, technically *EOM* does not work better or worse with limited angular range. The main output of *EOM* is a histogram of the *R*_g_ values (and sometimes a histogram of *D*_max_ values). *EOM* just characterizes the protein as compact or extended and provides a way of comparing the degree of compactness. The *R*_g_ and *D*_max_ information is contained in the lower angles. The high angle information is smeared due to polydispersity of the sample in case of a flexible protein. Nevertheless, we fully agree with your concerns about the *q* range and we did validation process through the SAXS data deposition on SASBDB. Through the review process in SASBDB, our data for mUbRab5_K116_, mUbRab5_K140_ and mUbRab5_K165_ have been successfully deposited to the SASBDB (ID: SASDCM6, SASDCN6 and SASDCP6, respectively). Therefore, we believe that the SAXS data we used are of sufficient quality for *EOM* modelling. Since the data will be released after acceptance on publication, you can only access them through the link below.

mUbRab5_K116_: https://www.sasbdb.org/data/SASDCM6/tkmwsdjl7l/

mUbRab5_K140_: https://www.sasbdb.org/data/SASDCN6/0ec25yspdo/

mUbRab5_K165_: https://www.sasbdb.org/data/SASDCP6/7wnrjfzgyv

No details are provided regarding the EOM modeling. For example, how many residues we're allowed to be flexible and what criteria where to determine which residues show be flexible? Do the results change if additional residues are allowed to be flexible? A narrow selected distribution near the high R_g_ and D_max_ tails of the pools in Figure 4 suggest that the number of flexible residues used to generate the pool might not be adequate to represent the flexibility in solution. Also unclear is how the linkage between Rab5 and ubiquitin is modeled. Does the EOM model disulfide linkages? If not, how was the linkage modeled? The details of the modeling are important and how they affect the results needs to be evaluated and discussed.

We added details for *EOM* analysis in the revised manuscript as described below. We believe that the preparation steps of input files for ubiquitinated proteins in *EOM* analysis are key steps, and it will be very useful to other scientists who wish to apply *EOM* to similar cases. The detailed description for *EOM* analysis is now described as follows:

(Subsection “Structural models of mUbRab5s derived by solution scattering”) “*EOM* was originally designed for multi–domain proteins with a flexible linker. Input files for *EOM* are a single linear protein sequence covering entire protein, atomic coordinates for each domain, and SAXS data (Tria et al., 2015). Because ubiquitinated proteins are not single polypeptides but branched ones, we generated the sequences and atomic coordinates of mUbRab5s as described in the Materials and methods. A pool of 10,000 independent models was generated based on the sequence and structural information from SAXS curve by *RANCH* (embedded in *EOM, ATSAS* package). Then, a genetic algorithm for the selection of an ensemble was performed by *GAJOE* with 100 times (embedded in *EOM, ATSAS* package). Finally, the best ensemble matched with the SAXS curve with lowest χ^2^ was selected."

(Subsection “Small angle X-ray scattering (SAXS) measurement and data processing”) “Since *EOM* accepted only single polypeptide chain with multiple domains, we generated the sequences and atomic coordinates of the mUbRab5s in the following four steps. […] Finally, we used the native isopeptide linkage for ensemble model calculations in *EOM* since a disulfide bond linkage is known to resemble the native ubiquitination linkage (Baker, Lewis, et al., 2013).”

"Interestingly, none of the EOM–derived structural models for mUbRab5s implicates any direct interaction between ubiquitin and Rab5" (subsection “Structural models of mUbRab5s derived by solution scattering”). Is EOM capable of producing models with direct interactions? Even if it could, what would it mean at the low resolution of SAXS?

We revised the sentence as follows:

(Subsection”Structural models of mUbRab5s derived by solution scattering”) “Similar to the molecular dynamics simulation of mUbRas (Baker, Lewis, et al., 2013), none of the ubiquitin molecules from the *EOM* models were located proximal to Rab5.”

Figure 4—figure supplement 1: Kratky–Debye and Porod–Debye plots indicate how extended a molecule is. Flexibility is inferred. Presumably the same results would be obtained from the calculated EOM curves that fit the data. The question is whether the SAXS data can distinguish flexible vs. extended. The interpretation should be more nuanced.

As the reviewer commented, it is very important to characterize a sample when performing SAXS experiments. Since any kinds of SAXS curve could be used for obtaining ab initio or ensemble model, one should be extremely careful in preparing the sample and initial data assessment. If the sample is homogenous and rigid, SAXS curve can be used for ab initio modeling. In contrast, if the sample is homogenous and flexible, ensemble analysis should be chosen for the method of choice. Judgement of flexibility of the sample is really important in the initial data assessment. From a SAXS curve, we can analyze whether the protein is folded or unfolded (extended) through traditional Kratky analysis (s vs. s^2^*I(s)). However, it was challenging to analysis of flexible versus unfolded protein using traditional Kratky analysis. Rambo and Tainer described a new analysis, called Porod–Debye law (s^4^ vs. s^4^*I(s)), for obtaining information of flexibility of a protein from a SAXS curve (Rambo & Tainer, 2011). Briefly, Porod–Debye plot can distinguish discrete conformational change (apo vs holo form of a protein with a substrate) and flexibility of the protein, while traditional Kratky analysis cannot. To obtain this information, one should take a look whether there is a plateau in Porod–Debye plot. If there is a plateau, it implies that the protein is compact or rigid, while loss of plateau reflects flexibility of the protein. The graphical explanation for this analysis is also well–presented in a website (http://www.bioisis.net/tutorial/12). In addition the Porod–Debye law has another advantage over Kratky analysis. While Kratky analysis requires relatively high angle data (0.01 < s < 0.3 Å^–1^), the Porod–Debye analysis can be done with similar low angular range as Guinier region. To make it easier for a reader to understand, we inserted the interpretation of flexibility analysis in our revised manuscript as follows:

(Subsection “Structural models of mUbRab5s derived by solution scattering”) “To check whether the ubiquitin moieties of Rab5 are flexible, we performed flexibility analysis using the Porod–Debye method (Rambo & Tainer, 2011). SAXS curves from the three mUbRab5s did not have a plateau in the Porod–Debye plot, implying that all of the mUbRab5s were flexible in solution (Figure 4—figure supplement 1).